# Sharp Empirical Bernstein Inequalities for the Variance of Bounded Random Variables

**Diego Martinez-Taboada** [1]   **Aaditya Ramdas** [1 2]

## Abstract

We develop novel "empirical Bernstein" inequalities for the variance of bounded random variables. Our inequalities hold under constant conditional variance and mean, without further assumptions like independence or identical distribution of the random variables, making them suitable for sequential decision making contexts. The results are instantiated for both the batch setting (where the sample size is fixed) and the sequential setting (where the sample size is a stopping time). Our bounds are asymptotically "sharp": when the data are iid, our CI adapts optimally to both unknown mean $\mu$ and unknown $\mathbb{V}[(X - \mu)^2]$, meaning that the first order term of our CI *exactly* matches that of the oracle Bernstein inequality which knows those quantities. We compare our results to a widely used (non-sharp) concentration inequality for the variance based on self-bounding random variables, showing both the theoretical gains and improved empirical performance of our approach. We finally extend our methods to work in any separable Hilbert space.

## 1. Introduction

Providing finite-sample confidence intervals for the variance of a random variable is a fundamental problem in statistical inference, as variance quantifies the dispersion of data and directly influences uncertainty in decision-making. Exact confidence intervals are particularly important because approximate methods can lead to misleading inferences, especially when sample sizes are small, data distributions are skewed, or underlying assumptions are violated.

[1]Department of Statistics & Data Science, Carnegie Mellon University, Pittsburgh, United States [2]Machine Learning Department, Carnegie Mellon University, Pittsburgh, United States. Correspondence to: Diego Martinez-Taboada <diegomar@andrew.cmu.edu>.

*Proceedings of the 43rd International Conference on Machine Learning*, Seoul, South Korea. PMLR 306, 2026. Copyright 2026 by the author(s).

In particular, exact confidence intervals for the *variance* play a central role in modern data-driven inference. Applications include multi-armed bandits (Audibert et al., 2009), off-policy evaluation (Thomas et al., 2015) and risk assessment (Huang et al., 2022), average treatment effect inference (Howard et al., 2021) and estimation (Neopane et al., 2025), sample compression (Maurer & Pontil, 2009), racing algorithms and boosting (Mnih et al., 2008), PAC-Bayes procedures (Tolstikhin & Seldin, 2013), and efficient confidence intervals (Austern & Mackey, 2022), among others. However, the inferential tools that are provided for the variance in all these previous works can be sharpened.

This contribution centers on the study of the variance of bounded random variables. Assume for the moment that we observe $X_1, \ldots, X_n$, which are independent and identically distributed to $X$, which is a random variable taking values on $[0, 1]$ with mean $\mu$ and variance $\mathbb{V}(X)$. The primary goal of this work is to derive confidence intervals for $\mathbb{V}(X)$ that both well work in practice and are asymptotically equivalent to those derived from the oracle Bernstein inequality. More specifically, we seek fully empirical confidence intervals whose width's leading term matches that of the $[0, 1]$-valued oracle Bernstein confidence interval for $\mathbb{V}(X)$. To recall, Bernstein's inequality for $\mathbb{V}(X)$ implies that $\mathbb{P}(\mathbb{V}(X) \in C_n) \geq 1 - \alpha$ with $C_n = [D_n - R_n, D_n + R_n]$, where $D_n$ is the center of the confidence interval and $R_n$ is the radius

$$R_n = \sqrt{\frac{2\mathbb{V}[(X_i - \mu)^2]\log(2/\alpha)}{n}} + \frac{\log(2/\alpha)}{3n}.$$

The challenge in constructing the above interval in practice is that both $\mu$ and $2\mathbb{V}\left[(X_i - \mu)^2\right]$ are unknown. A fully empirical confidence interval is called *sharp* if its width asymptotically matches that of the first term above, including constants. To elaborate, when estimating the mean, Bernstein inequalities (Bernstein, 1927; Bennett, 1962) are widely known for leading to closed-form, tight confidence intervals. However, their practicality is limited, as they require knowing a bound on the variance of the random variables (that is better than the trivial bound implied by the bounds on the random variables). For this reason, they establish a natural "oracle" benchmark for fully empirical confidence sets that only exploit knowledge on the bound of the random variables.

Furthermore, some of the aforementioned applications rely on confidence sequences, which are anytime-valid counterparts of confidence intervals. A $(1 - \alpha)$-confidence interval $C_{\mathrm{CI}}$ for a target parameter $\theta$ is a random set such that $P(\theta \in C_{\mathrm{CI}}) \geq 1 - \alpha$, where $C_{\mathrm{CI}}$ is built after having observed a fixed number of observations. In contrast, $(C_t)_{t \geq 1}$ is a $(1 - \alpha)$-confidence sequence if $P(\forall t \geq 1 : \theta \in C_t) \geq 1 - \alpha$, with $t$ representing the number of observations collected sequentially. Confidence sequences are of key importance in online settings, where data is observed sequentially and probabilistic guarantees that hold at stopping times are often desired: confidence sequences allow for sequential procedures that are continuously monitored and adaptively adjusted. For instance, the optimal adaptive Neyman allocation in causal inference (Neopane et al., 2025) is based on confidence sequences for the variance.

Variance confidence sequences are particularly valuable when uncertainty quantification must be performed sequentially and in a data-dependent manner. For instance, the optimal adaptive Neyman allocation in causal inference (Neopane et al., 2025) is based on confidence sequences for the variance. Similarly, in online learning and bandit problems, high-probability regret guarantees are often derived via concentration arguments, and sharp variance bounds enable tighter such conversions (Mnih et al., 2008; Audibert et al., 2009). Furthermore, in risk-sensitive decision-making, such as safe reinforcement learning or clinical trials (Kazerouni et al., 2017; Howard et al., 2021), sequential variance bounds allow practitioners to monitor and constrain risk in real time without relying on fixed horizons. More broadly, empirical Bernstein-type bounds play an important role in adaptive stopping rules, best-arm identification, and Monte Carlo estimation, where tighter variance estimates directly translate into improved sample efficiency.

In many such online settings, the assumption that data points are independent and identically distributed (iid) is often too strong and unrealistic due to the dynamic and evolving nature of data streams. Unlike traditional offline analyses where data can be assumed to come from a fixed distribution, online environments involve sequentially arriving data that may exhibit temporal dependencies. Thus, we seek to develop concentration inequalities that only require the following assumption to hold (where $\mathbb{E}_t$ and $\mathbb{V}_t$ denote conditional expectations and variances, respectively; these definitions are later formalized in Section 3).

**Assumption 1.1.** The stream of random variables $X_1, X_2, \ldots$ is such that

$$X_t \in [0, 1], \quad \mathbb{E}_{t-1} X_t = \mu, \quad \mathbb{V}_{t-1} X_t = \sigma^2.$$

Note that any bounded random variable can be rescaled to belong to $[0, 1]$, and so the first of the conditions can be assumed without loss of generality in the bounded setting.

Since boundedness is a prerequisite for existing empirical Bernstein inequalities concerning the mean, it is unavoidable here too. Other than boundedness, Assumption 1.1 remains substantially weak, replacing the traditional i.i.d. assumption with a broader martingale dependence structure. Furthermore, the conditional constant mean and variance are arguably the least we may assume if we wish to estimate "the variance". Since all bounded i.i.d. sequences can be rescaled to satisfy Assumption 1.1, our framework constitutes a strictly more general approach than the standard bounded i.i.d. assumptions prevalent in the literature. In particular, Assumption 1.1 is attained in all aforementioned applications, with some of them requiring i.i.d. data (Maurer & Pontil, 2009; Austern & Mackey, 2022), and others requiring only martingale dependence (Howard et al., 2021; Neopane et al., 2025).

We study the problem of providing confidence intervals and sequences for the variance of bounded random variables under Assumption 1.1. Our goal is to derive a fully data-dependent, nonasymptotically valid confidence interval for $\sigma^2$, without knowing either the mean or the fourth moment, while also being asymptotically as sharp as the oracle Bernstein bound that knows those quantities. That is, we want nonasymptotic validity and asymptotic sharpness for an explicit fully-data dependent confidence interval. Our contributions are three-fold:

- We provide novel confidence sequences for the variance (Corollary 4.3 and Corollary 4.4), that are derived from novel supermartingale constructions (Theorem 4.1 and Corollary 4.2). We instantiate the results for the sequential setting in Section 4.2 and Section 4.3, and for the batch setting (where the confidence sequence reduces to a confidence interval) in Section 4.4. Confidence sequences for the standard deviation (std) $\sigma$ can also be immediately derived by taking the square root of the confidence sequences for the variance.

- Theoretically, we prove the sharpness of our inequalities by showing that the first order term of the novel confidence interval exactly matches that of the oracle Bernstein inequality (Corollary 4.8).

- Empirically, we illustrate how our proposed inequalities substantially outperform those of Maurer & Pontil (2009, Theorem 10) in Section 5, which constitute the existing sharpest inequalities for the standard deviation, to the best of our knowledge. We further illustrate how our confidence sequences improve the adaptive Neyman allocation procedure from Neopane et al. (2025), which requires anytime-valid inference for the variance of the potential outcomes, but used inferior ones to ours.

## 2. Related Work

**Current concentration inequalities for the variance.** Upper and lower inequalities for the variance were presented in Maurer & Pontil (2009, Theorem 10). They are based on the concentration of self-bounding random variables (Maurer, 2006). Another concentration inequality for the variance can be found in the proof of Audibert et al. (2009, Theorem 1), which decouples the analysis into those of the mean and second centered moment, in a similar spirit to our contribution. However, these inequalities rely on conservatively upper bounding the *variance of the empirical variance*, thus being loose (this is also the case for Voráček & Orabona (2025, Theorem 12)). In contrast, our inequalities empirically estimate the variance of the empirical variance, resulting in tighter confidence sets. We defer an extended comparison of these inequalities to Section 4.4. Less closely related to our work, other inequalities rely on the Kolmogorov-Smirnov distance between the empirical distribution and a reference distribution.

**Empirical Bernstein inequalities for the mean.** Based on combining Bennett's inequality and upper concentration inequalities for the variance, Maurer & Pontil (2009, Theorem 11) proposed a well known empirical Bernstein inequality for the mean, improving a similar inequality presented in Audibert et al. (2009, Theorem 1). These inequalities are not asymptotically sharp as presented (in that the first order limiting width does not match that of the oracle Bernstein inequality, including constants). However, they can be amended to recover sharpness if the probability split of the union bound is carefully designed (as pointed out recently in Wang & Ramdas (2025, Section B.2)). Nevertheless, their bounds were empirically significantly looser (Waudby-Smith & Ramdas, 2024, Figure 3) than those presented in Howard et al. (2021, Theorem 4) and Waudby-Smith & Ramdas (2024, Theorem 2), which are also sharp. Related contributions include Mnih et al. (2008, Theorem 2), Jang et al. (2023, Corollary 4), Orabona & Jun (2023, Theorem 3), and Martinez-Taboada & Ramdas (2026, Corollary 1).

**Time-uniform Chernoff inequalities.** Our work falls under the time-uniform Chernoff inequalities umbrella from Howard et al. (2020; 2021); Waudby-Smith & Ramdas (2024). A key proof technique of this line of work is the derivation of sophisticated nonnegative supermartingales, followed by an application of Ville's inequality (Ville, 1939), an anytime-valid version of Markov's inequality.

## 3. Background

Let us start by presenting the concepts of *filtration* and *supermartingale*, which will be heavily exploited in this work to go beyond the iid setting. Consider a filtered measurable space $(\Omega, \mathcal{F})$, where the filtration $\mathcal{F} = (\mathcal{F}_t)_{t \geq 0}$

is a sequence of $\sigma$-algebras such that $\mathcal{F}_t \subseteq \mathcal{F}_{t+1}$, $t \geq 0$. The canonical filtration $\mathcal{F}_t = \sigma(X_1, \ldots, X_t)$, with $\mathcal{F}_0$ being trivial, is considered throughout. A stochastic process $M \equiv (M_t)_{t \geq 0}$ is a sequence of random variables that are adapted to $(\mathcal{F}_t)_{t \geq 0}$, i.e., $M_t$ is $\mathcal{F}_t$-measurable for all $t$. $M$ is called *predictable* if $M_t$ is $\mathcal{F}_{t-1}$-measurable for all $t$. An integrable stochastic process $M$ is a supermartingale if $\mathbb{E}[M_{t+1}|\mathcal{F}_t] \leq M_t$ for all $t$. We use $\mathbb{E}_t[\cdot]$ and $\mathbb{V}_t[\cdot]$ in short for $\mathbb{E}[\cdot|\mathcal{F}_t]$ and $\mathbb{V}[\cdot|\mathcal{F}_t]$, respectively. Inequalities between random variables are always interpreted to hold almost surely.

As exhibited in later sections, our concentration inequalities will be derived as Chernoff inequalities. In contrast to more classical inequalities, our results come with anytime validity (that is, they hold at any stopping time), derived using the following anytime-valid version of Markov's inequality.

**Theorem 3.1** (Ville's inequality). *For any nonnegative supermartingale $(M_t)_{t \geq 0}$ and $x > 0$,*

$$\mathbb{P}\left(\exists t \geq 0 : M_t \geq x\right) \leq \frac{\mathbb{E}M_0}{x}.$$

Powerful nonnegative supermartingale constructions are usually at the heart of anytime valid concentration inequalities. For example, the following sharp empirical Bernstein inequality from Howard et al. (2021) and Waudby-Smith & Ramdas (2024) is derived from a nonnegative supermartingale.

**Theorem 3.2** (Empirical Bernstein inequality). *Let $X_1, X_2, \ldots$ be a stream of random variables such that, for all $t \geq 1$, it holds that $X_t \in [0,1]$ and $\mathbb{E}_{t-1} X_t = \mu$. Let $\psi_E(\lambda) = -\log \lambda - \lambda$. For any $[0,1)$-valued predictable sequence $(\lambda_i)_{i \geq 1}$ such that $\lambda_1 > 0$, it holds that*

$$\left(\frac{\sum_{i=1}^n \lambda_i X_i}{\sum_{i=1}^n \lambda_i} \pm \frac{\log\left(\frac{2}{\delta}\right) + \sum_{i=1}^n \psi_E(\lambda_i)\left(X_i - \hat{\mu}_{i-1}\right)^2}{\sum_{i=1}^n \lambda_i}\right)$$

*is a $1 - \delta$ confidence sequence for $\mu$.*

We will modify Theorem 3.2 in later sections in order to derive our results. The sequence $(\lambda_i)_{i \geq 1}$ is referred to as 'predictable plug-ins'. They play the role of the parameter $\lambda$ that naturally appears in all the Chernoff inequality derivations; nevertheless, instead of they being equal for each $i$ and theoretically optimized, they are empirically and sequentially chosen. The choice of the predictable plug-ins is key in the performance of the inequalities, and will be discussed throughout our work. Besides making use of predictable plug-ins in empirical Bernstein-type supermartingales, we will also exploit them in the following anytime valid version of Bennett's inequality.

**Theorem 3.3** (Anytime valid Bennett's inequality). *Let $X_1, X_2, \ldots$ be a stream of random variables such that, for*

*all $t \geq 1$, it holds that $X_t \in [0,1]$, $\mathbb{E}_{t-1} X_t = \mu$, and $\mathbb{V}_{t-1} X_t = \sigma^2$. Let $\psi_P(\lambda) = \exp(\lambda) - \lambda - 1$. For any $\mathbb{R}_+$-valued predictable sequence $(\tilde{\lambda}_i)_{i \geq 1}$, it holds that*

$$\left( \frac{\sum_{i \leq t} \tilde{\lambda}_i X_i}{\sum_{i \leq t} \tilde{\lambda}_i} \pm \frac{\log(2/\delta) + \sigma^2 \sum_{i \leq t} \psi_P(\tilde{\lambda}_i)}{\sum_{i \leq t} \tilde{\lambda}_i} \right)$$

*is a $1 - \delta$ confidence sequence for $\mu$.*

While this result is technically novel (and so we present a proof in Appendix D.1), it can be derived using the techniques in (Howard et al., 2020; Waudby-Smith & Ramdas, 2024). It would generally lack any practical use, given that $\sigma$ is typically unknown. Nonetheless, we will invoke it in combination with an empirical Bernstein inequality for $\sigma^2$, thus making it actionable.

# 4. Main Results

We are now ready to derive novel confidence intervals and sequences for the variance of bounded random variables under Assumption 1.1. Two natural methodological strategies arise. The first constructs confidence intervals by applying concentration inequalities for the variance to an empirical variance estimator, itself centered around an empirical estimator for the mean. For lower confidence intervals, the concentration guarantee cannot be invoked directly unless we also account for uncertainty in the empirical mean. This is the approach adopted in this work. Motivated by the strong empirical performance of the empirical Bernstein inequalities for the mean of bounded data, we develop another of these inequalities to control the empirical variance. Remarkably, for lower confidence intervals we demonstrate that empirical concentration inequalities for the mean are unnecessary: we can construct confidence intervals for the mean that scale linearly with the the square root of the true variance (without needing to estimate it) by solving a quadratic equation. Further details follow in the subsequent exposition.

The second strategy is to express the variance as $\sigma^2 = E_{t-1} X_t^2 - \mu^2$, where by Assumption 1.1 the conditional expectation remains constant across time. Moreover, since $X_t \in [0,1]$, the squared observations also lie in the unit interval, allowing the application of concentration inequalities for bounded variables to both the mean and the second moment; these bounds can then be combined via the union bound. However, as demonstrated in Appendix F.1, this strategy yields inferior theoretical and empirical performance relative to the first approach. Intuitively, the performance gap stems from $\mathbb{V}(X - \mu)^2 \leq \mathbb{V} X^2$ (our method achieves the smaller variance).

The section is organized as follows. In Section 4.1, we present the theoretical foundation of all the inequalities

derived thereafter, namely a novel nonnegative supermartingale construction and its corollary. Section 4.2 and Section 4.3 make use of such theoretical tools to derive upper and lower confidence sequences, respectively. In particular, the latter (Section 4.3) makes essential use of an auxiliary Bennett-type concentration inequality for the estimator of the mean. Section 4.4 instantiates such confidence sequences in the (more classical) batch setting, where they reduce to confidence intervals. We defer an extension of the results to Hilbert spaces to Appendix A.

## 4.1. A Nonnegative Supermartingale Construction

We begin by introducing two nonnegative supermartingale constructions that serve as the theoretical foundation for the inequalities derived in this work; these nonnegative supermartingales will lead to concentration bounds when in conjunction with Ville's inequality. Its proof may be found in Appendix D.2.

**Theorem 4.1.** *Let Assumption 1.1 hold. For a $[0,1]$-valued predictable sequence $(\widehat{\mu}_i)_{i \geq 1}$, denote*

$$\tilde{\sigma}_i^2 = \sigma^2 + (\widehat{\mu}_i - \mu)^2.$$

*For any $[0,1]$-valued predictable sequence $(\widehat{\sigma}_i)_{i \geq 1}$ and any $[0,1)$-valued predictable sequence $(\lambda_i)_{i \geq 1}$, the processes $(S_t^+)_{t \geq 0}$ and $(S_t^-)_{t \geq 0}$, with $S_0^+ := 1$, and $S_0^- := 1$, and*

$$S_t^{\pm} := \exp \left\{ \sum_{i \leq t} \pm \lambda_i \left[ (X_i - \widehat{\mu}_i)^2 - \tilde{\sigma}_i^2 \right] \right.$$
$$\left. - \psi_E(\lambda_i) \left[ (X_i - \widehat{\mu}_i)^2 - \widehat{\sigma}_i^2 \right]^2 \right\}, \quad t \geq 1,$$

*are nonnegative supermartingales.*

Theorem 4.1 modifies the supermartingales that give way to Theorem 3.2. However, in contrast to Theorem 3.2, the conditional means of the random variables $(X_i - \widehat{\mu}_i)^2$ under study are not constant. For this reason, the analysis is more convoluted in our setting. Hence, in order to provide concentration results for the variance, we denote

$$R_{t,\alpha} := \frac{\log(1/\alpha) + \sum_{i \leq t} \psi_E(\lambda_i) \left( (X_i - \widehat{\mu}_i)^2 - \widehat{\sigma}_i^2 \right)^2}{\sum_{i \leq t} \lambda_i},$$

$$D_t := \frac{\sum_{i \leq t} \lambda_i (X_i - \widehat{\mu}_i)^2}{\sum_{i \leq t} \lambda_i}, \quad E_t := \frac{\sum_{i \leq t} \lambda_i (\widehat{\mu}_i - \mu)^2}{\sum_{i \leq t} \lambda_i},$$

where $D_t$ and $R_{t,\alpha}$ will represent the center and radius of the confidence interval (respectively), and $E_t$ an extra term that appears due to the mean estimator error. The following corollary is a direct consequence of Theorem 4.1, as a result of applying Ville's inequality to the nonnegative supermartingales. We defer its proof to Appendix D.3.

**Corollary 4.2.** *Let Assumption 1.1 hold. For any $[0, 1)$-valued predictable sequence $(\lambda_i)_{i\geq 1}$ and any $[0, 1]$-valued predictable sequences $(\widehat{\mu}_i)_{i\geq 1}$ and $(\widehat{\sigma}_i)_{i\geq 1}$, it holds that*

$$\left(D_t - E_t \pm R_{t,\frac{\alpha}{2}}\right)$$

*is a $1 - \alpha$ confidence sequence for $\sigma^2$.*

The confidence sequence provided by Corollary 4.2 cannot be invoked in practice: since $\mu$ is unknown, $E_t$ is also unknown.

## 4.2. Upper Confidence Sequence for the Variance

In spite of $E_t$ being unknown, this term poses no challenge for the upper confidence sequence, as we can simply lower bound it by 0. That is, if $[0, D_t - E_t + R_{t,\alpha})$ is an $\alpha$-level upper confidence sequence for the variance, so is $[0, D_t + R_{t,\alpha})$, given that $E_t$ is nonnegative. We formalize such an observation in the following corollary.

**Corollary 4.3** (Upper empirical Bernstein for the variance). *Let Assumption 1.1 hold. For any $[0, 1]$-valued predictable sequences $(\widehat{\mu}_i)_{i\geq 1}$ and $(\widehat{\sigma}_i)_{i\geq 1}$, and any $[0, 1)$-valued predictable sequence $(\lambda_i)_{i\geq 1}$, it holds that $[0, U_t)$ is a $1 - \alpha$ upper confidence sequence for $\sigma^2$, where*

$$U_t = D_t + R_{t,\alpha}.$$

It remains to discuss the choice of predictable sequences. We suggest to take

$$\widehat{\sigma}_t^2 := \frac{c_3 + \sum_{i\leq t-1}(X_i - \bar{\mu}_i)^2}{t}, \quad \bar{\mu}_t := \frac{c_4 + \sum_{i\leq t-1} X_i}{t},$$

where $c_3, c_4 \in [0, 1]$ are constant, as well as $\widehat{\mu}_t = \bar{\mu}_t$. These specific choices are only proposed due to their computational simplicity; but our upper bound holds for any other choice of mean and variance estimator.

Following the discussion from Waudby-Smith & Ramdas (2024, Section 3.3) for confidence sequences, we propose to take the predictable plug-ins

$$\lambda_{t,u,\alpha}^{\mathrm{CS}} := \sqrt{\frac{2\log(1/\alpha)}{\widehat{m}_{4,t}^2 t \log(1+t)}} \wedge c_1$$

where

$$\widehat{m}_{4,t}^2 := \frac{c_2 + \sum_{i\leq t-1}\left[(X_i - \widehat{\mu}_i)^2 - \widehat{\sigma}_i^2\right]^2}{t},$$

with $c_1 \in (0, 1)$, and $c_2 \in [0, 1]$. Reasonable defaults are $c_1 = \frac{1}{2}$, $c_2 = \frac{1}{2^4}$, $c_3 = \frac{1}{2^2}$, and $c_4 = \frac{1}{2}$. The values $c_2 - c_4$ regularize the mean and variance estimators, so their impact decays considerably fast. Similarly to Waudby-Smith & Ramdas (2024), the constant $c_1$ prevents the predictable sequence from exploding, and the performance is not highly affected by choices that are not too close to 1.

## 4.3. Lower Confidence Sequence for the Variance

In order to provide a lower confidence sequence, we must control the term $E_t$, which depends on the terms $|\mu - \widehat{\mu}_i|$, with $i \leq t$. This can be done if $(\widehat{\mu}_t)_{t\geq 1}$ is such that a confidence sequence for $|\widehat{\mu}_i - \mu|$ can be provided (we ought to use confidence sequences instead of confidence intervals in order to avoid union bounding over all $i \leq t$). If $|\widehat{\mu}_i - \mu| \leq \tilde{R}_{i,\alpha_1}$ for all $i \geq 1$ with probability $1 - \alpha_1$, then

$$D_t - \frac{\sum_{i\leq t}\lambda_i \tilde{R}_{i,\alpha_1}^2}{\sum_{i\leq t}\lambda_i} - R_{t,\alpha_2} \tag{1}$$

yields a $(1 - \alpha)$-lower confidence sequence for $\sigma^2$ with $\alpha_1 + \alpha_2 = \alpha$.

Naturally, we aim for $\tilde{R}_{i,\alpha_1}$ to be as small as possible. Given the strong practical performance of empirical Bernstein inequalities, one might initially consider selecting yet another such inequality for $\mu$ in order to derive $\tilde{R}_{i,\alpha_1}$. However, a better approach is available. If $\tilde{R}_{i,\alpha_1}$ depends linearly on $\sigma$, we can obtain a closed-form confidence interval for $\sigma$ by solving a quadratic equation. In particular, we may combine the empirical mean in conjunction with the oracle Bennett inequality to yield closed-form confidence intervals. Although alternative options exist, such as Lee & Valiant (2022, Theorem 1), we favor the theoretical Bennett inequality because it yields explicit and small constants (unlike e.g. the aforementioned alternative). Furthermore, the theoretical Bennett inequality used holds with anytime validity and under martingale dependence. We also deliberately avoid betting-based approaches, as our goal is to obtain closed-form confidence intervals.

In particular, we propose to obtain $\tilde{R}_{i,\delta}$ based on the anytime valid Bennett's inequality presented in Theorem 3.3. That is, take

$$\widehat{\mu}_t = \frac{\sum_{i=1}^{t-1}\tilde{\lambda}_i X_i}{\sum_{i=1}^{t-1}\tilde{\lambda}_i}, \; \tilde{R}_{t,\alpha_1} = \frac{\log(2/\alpha_1) + \sigma^2\sum_{i=1}^{t-1}\psi_P(\tilde{\lambda}_i)}{\sum_{i=1}^{t-1}\tilde{\lambda}_i}, \tag{2}$$

for $t \geq 2$, as well as $\widehat{\mu}_1 = \frac{1}{2}$ and $\tilde{R}_{1,\alpha_1} = \frac{1}{2}$. Substituting (2) in (1) leads to a quadratic polynomial on $\sigma^2$. Equaling $\sigma^2$ to such a polynomial and solving for $\sigma^2$ yields our lower confidence sequence. In order to formalize this, denote

$$\tilde{C}_t^{(2)} := \frac{\left(\sum_{i=1}^{t-1}\psi_P(\tilde{\lambda}_i)\right)^2}{\left(\sum_{i=1}^{t-1}\tilde{\lambda}_i\right)^2}, \quad \tilde{C}_{t,\delta}^{(0)} := \frac{\log^2(2/\delta)}{\left(\sum_{i=1}^{t-1}\tilde{\lambda}_i\right)^2},$$

$$\tilde{C}_{t,\delta}^{(1)} := \frac{2\log(2/\delta)\sum_{i=1}^{t-1}\psi_P(\tilde{\lambda}_i)}{\left(\sum_{i=1}^{t-1}\tilde{\lambda}_i\right)^2},$$

as well as

$$C_t^{(2)} := \frac{\sum_{i \leq t} \lambda_i \tilde{C}_i^{(2)}}{\sum_{i \leq t} \lambda_i}, \quad C_{t,\delta}^{(0)} := \frac{\sum_{i \leq t} \lambda_i \tilde{C}_{i,\delta}}{\sum_{i \leq t} \lambda_i},$$

$$C_{t,\delta}^{(1)} := 1 + \frac{\sum_{i \leq t} \lambda_i \tilde{B}_{i,\delta}}{\sum_{i \leq t} \lambda_i}$$

(the numbers represent the degrees in the quadratic polynomial). Under this notation, we are ready to present Corollary 4.4, a lower confidence sequence for the variance. Its proof has been deferred to Appendix D.4.

**Corollary 4.4** (Lower empirical Bernstein for the variance). *Let Assumption 1.1 hold. For $(\widehat{\mu}_i)_{i \geq 1}$ defined as in (2), any $[0,1]$-valued predictable sequence $(\widehat{\sigma}_i^2)_{i \geq 1}$, any $[0,1)$-valued predictable sequence $(\lambda_i)_{i \geq 1}$, and any $[0,\infty)$-valued predictable sequence $(\tilde{\lambda}_i)_{i \geq 1}$, it holds that $(L_t, \infty)$ is a $1 - \alpha$ lower confidence sequence for $\sigma^2$, where $\alpha_1 + \alpha_2 = \alpha$ and $L_t$ equals*

$$\frac{-C_{t,\alpha_1}^{(1)} + \sqrt{\left(C_{t,\alpha_1}^{(1)}\right)^2 + 4C_t^{(2)}(D_t - C_{t,\alpha_1}^{(0)} - R_{t,\alpha_2})}}{2C_t^{(2)}}. \tag{3}$$

It remains to discuss the choice of predictable plug-ins. Analogously to the upper inequality plug-ins, it would be natural to take $\lambda_{t,l,\alpha_2}^{\mathrm{CS}} = \lambda_{t,u,\alpha_2}^{\mathrm{CS}}$. However, the lower inequality includes the extra terms $\tilde{R}_{i,\alpha_1}$ that ought to be accounted for. Taking $\lambda_i > 0$ for $i$ such that $\tilde{R}_{i,\alpha_1} > 1$ would add a summand that is vacuous.[1] For this reason, we propose to take $\lambda_{t,l,\alpha_2}^{\mathrm{CS}} := \lambda_{t,u,\alpha_2}^{\mathrm{CS}}$ if $t \geq 2$ and $\frac{\log(2/\alpha_1)+\hat{\sigma}_t^2 \sum_{i=1}^{t-1} \psi_P(\tilde{\lambda}_i)}{\sum_{i=1}^{t-1} \tilde{\lambda}_i} \leq 1$, and $\lambda_{t,l,\alpha_2}^{\mathrm{CS}} := 0$ otherwise.

Note that the threshold is only an approximation of $\tilde{R}_{i,\alpha_1}$, given that the latter is unknown in practice. Seeking a confidence sequence for $\mu$, we propose to take $(\tilde{\lambda}_i)_{i \geq 1}$ as

$$\tilde{\lambda}_t := \sqrt{\frac{2\log(2/\alpha)}{\widehat{\sigma}_t^2 t \log(1+t)}} \wedge c_5, \tag{4}$$

with $c_5$ being a constant in $(0, \infty)$, with a sensible default being $c_5 = 2$. The choice of the split of $\alpha$ into $\alpha_1$ and $\alpha_2$ is also of importance. In the next section, we analyze specific splits for retrieving optimal asymptotical behavior. If having access to artificial samples similarly distributed to the random variables under study, the probability split can be optimized for using those observations as a reference. However, the split split $\alpha_1 = \alpha_2 = \frac{\alpha}{2}$ works generally well in practice.

---

[1] It would also be reasonable to take the minimum of $\tilde{R}_{i,\alpha_1}$ and 1. We explore this alternative in Appendix F.

## 4.4. Upper and Lower Confidence Intervals

In the more classical batch setting, we observe a fixed number of observations $X_1, \ldots, X_n$, with $n$ known in advance. Given that confidence sequences are, in particular, confidence intervals for a fixed $t = n$, both Corollary 4.3 and Corollary 4.4 immediately establish confidence intervals. However, the choice of predictable plug-ins used in such corollaries should now be driven by minimizing the expected interval width at a specific $t \equiv n$, rather than being tight uniformly over $t$.

For this reason, in order to optimize the upper confidence interval for a fixed $t \equiv n$, we take

$$\lambda_{i,u,\alpha}^{\mathrm{CI}} := \sqrt{\frac{2\log(1/\alpha)}{\widehat{m}_{4,i}^2 n}} \wedge c_1. \tag{5}$$

Following the same line of reasoning as in Section 4.3, the plug-ins for the lower confidence intervals are defined as a slight modification of those for the upper confidence sequence. Accordingly, we take $\lambda_{t,l,\alpha_2}^{\mathrm{CI}} := \lambda_{t,u,\alpha_2}^{\mathrm{CI}}$ if $t \geq 2$ and $\frac{\log(2/\alpha_1)+\hat{\sigma}_t^2 \sum_{i=1}^{t-1} \psi_P(\tilde{\lambda}_i)}{\sum_{i=1}^{t-1} \tilde{\lambda}_i} \leq 1$, and $\lambda_{t,l,\alpha_2}^{\mathrm{CI}}$ otherwise. The remaining parameters and estimators are defined in the same manner as in the preceding sections.

### AN ANALYSIS OF THE WIDTHS FOR THE VARIANCE

In order to draw comparisons with related inequalities, we analyze the asymptotic first order term of the novel confidence intervals for the variance. As emphasized in Section 1, in the event of $(X_i - \mu)^2$ having constant conditional variance, the benchmark for the first order terms are those of oracle Bernstein confidence intervals, i.e. $\sqrt{2\mathbb{V}\left[(X_i - \mu)^2\right]\log(1/\alpha)}$. That is, we seek to prove that both $\sqrt{n}(U_n - D_n)$ and $\sqrt{n}(D_n - L_n)$ converge almost surely to such quantity. Accordingly, we make the following assumption throughout.

**Assumption 4.5.** $X_1, X_2, \ldots$ is such that $\mathbb{V}_{i-1}\left[(X_i - \mu)^2\right]$ is constant across $i$.

We highlight that the concentration inequalities presented in this paper do not require Assumption 4.5 to be valid. We only impose Assumption 4.5 to compare the limiting width of our empirical inequalities to those of the oracle Bernstein inequality.

We will implicitly also assume that the predictable sequences $(\widehat{\mu}_i)_{i \in [n]}$ and $(\widehat{\sigma}_i^2)_{i \in [n]}$ are defined as in Section 4.2 or Section 4.3, with $c_2 \wedge c_3 > 0$. The condition $c_2 \wedge c_3 > 0$ is not necessary for the proofs to hold, but they follow cleaner with it.

For simplicity, we will focus on the asymptotic behavior of

both

$$\sqrt{n}R_{n,\alpha}, \quad \sqrt{n}\left(\frac{\sum_{i\leq t}\lambda_{i,\alpha_{2,n}}\tilde{R}_{i,\alpha_{1,n}}^2}{\sum_{i\leq t}\lambda_{i,\alpha_{2,n}}} + R_{n,\alpha_{2,n}}\right),$$

where we define $\lambda_{i,\alpha_{2,n}} = \lambda_{t,u,\alpha_{2,n}}^{\mathrm{CI}}$. These two quantities correspond to the first order widths of the upper and lower confidence intervals above and below the estimate $D_t$, respectively, if taking the plug-ins $\lambda_{i,\alpha_{2,n}}$.[2] Note that, in contrast to the previous section, we emphasize the dependence of the split $\alpha = \alpha_{1,n} + \alpha_{2,n}$ on $n$, which we will exploit to recover optimal first order terms.

We decouple the analysis in two parts, one involving the $R_i$'s and the other involving the $\tilde{R}_i$'s. We start by establishing that the former converges almost surely to the oracle Bernstein first order term for the right choices of $\alpha_{2,n}$. The proof can be found in Appendix D.5.

**Theorem 4.6.** *Let $(\delta_n)_{n\geq 1}$ be a deterministic sequence such that $\delta_n > 0$ and $\delta_n \nearrow \delta > 0$. If Assumption 1.1 and Assumption 4.5 hold, then*

$$\sqrt{n}R_{n,\delta_n} \stackrel{a.s.}{\to} \sqrt{2\mathbb{V}\left[(X_i - \mu)^2\right]\log(1/\delta)}.$$

Second, we prove that the extra term that appears in the lower confidence interval converges to zero almost surely for right choices of $\alpha_{1,n}$. The proof can be found in Appendix D.6.

**Theorem 4.7.** *Let $\alpha = \alpha_{1,n} + \alpha_{2,n}$ be such that $\alpha_{1,n} = \Omega\left(\frac{1}{\log(n)}\right)$ and $\alpha_{2,n} \to \alpha$. If Assumption 1.1 and Assumption 4.5 hold, then*

$$\sqrt{n}\frac{\sum_{i\leq t}\lambda_{i,\alpha_{2,n}}\tilde{R}_{i,\alpha_{1,n}}^2}{\sum_{i\leq t}\lambda_{i,\alpha_{2,n}}} \stackrel{a.s.}{\to} 0.$$

Taking $\delta_n = \alpha$, it immediately follows from Theorem 4.6 that the upper confidence interval's first order term is asymptotically almost surely equal to that of the oracle Bernstein confidence interval. To derive the analogous conclusion for the lower confidence interval, it suffices to take

$$\alpha_{1,n} = \frac{1}{\log n}\alpha, \quad \delta_n = \alpha_{2,n} = \frac{\log(n)-1}{\log(n)}\alpha \quad (6)$$

in Theorem 4.7 and Theorem 4.6, respectively. These claims are formalized in the following corollary.

**Corollary 4.8** (Sharpness). *Let the predictable sequences $(\widehat{\mu}_i)_{i\in[n]}$ and $(\widehat{\sigma}_i^2)_{i\in[n]}$ be defined as in Section 4.2 or Section 4.3, with $c_2 \wedge c_3 > 0$. Let Assumption 1.1 and Assumption 4.5 hold. If $\alpha = \alpha_{1,n} + \alpha_{2,n}$ as defined in (6),*

---

[2]Note that the plug-ins proposed in Section 4.4 are slightly different (they may take the value 0 for small enough $t$). However, the proofs can be easily extended to $\lambda_{t,l,\alpha_2}^{\mathrm{CI}}$ after realizing that these two plug-ins are equal almost surely for big enough $t$; we believe the simplification is convenient for ease of presentation.

*then*

$$\sqrt{n}(U_n - D_n) \stackrel{a.s.}{\to} \sqrt{2\mathbb{V}\left[(X_i - \mu)^2\right]\log(1/\alpha)},$$
$$\sqrt{n}(D_n - L_n) \stackrel{a.s.}{\to} \sqrt{2\mathbb{V}\left[(X_i - \mu)^2\right]\log(1/\alpha)}.$$

## COMPARISON TO EXISTING INEQUALITIES

We compare our proposed inequalities to those in Audibert et al. (2009) and Maurer & Pontil (2009). As mentioned in Section 2, upper and lower inequalities for the variance were previously established in the work of Audibert et al. (2009, Theorem 1) and Maurer & Pontil (2009, Theorem 10). Both these inequalities make use of $\mathbb{V}[(X_i - \mu)^2] \leq \sigma^2$, directly or indirectly. While Audibert et al. (2009, Theorem 1) makes direct use of it by leveraging Bernstein-type inequalities, Maurer & Pontil (2009, Theorem 10) makes indirectly use of it through self-bounding arguments. Maurer & Pontil (2009, Theorem 10), which is the sharper of the two, yields the $(1 - \alpha)$-confidence interval

$$\sigma \in \left(\sqrt{V_n} \pm \sqrt{\frac{2\log(2/\alpha)}{n-1}}\right),$$

where $V_n$ is the empirical variance. That is,

$$\sigma^2 \in \left(V_n + \frac{2\log(2/\alpha)}{n-1} \pm 2\sqrt{V_n\frac{2\log(2/\alpha)}{n-1}}\right).$$

In view of $V_n \to \sigma^2$ almost surely, we observe that the radii of these two-sided confidence intervals scaled by $\sqrt{n}$ are roughly

$$2\sigma\sqrt{2\log(2/\alpha)},$$

which is larger the limiting first order width of our confidence intervals (Corollary 4.8) in view of $1 < 2$ and

$$\mathbb{V}\left[(X_i - \mu)^2\right] \leq \mathbb{E}\left[(X_i - \mu)^2\right] - \mathbb{E}^2\left[(X_i - \mu)^2\right]$$
$$= \sigma^2(1 - \sigma^2) \leq \sigma^2.$$

This theoretical edge also manifests in practice, with the empirical widths of our inequalities proving considerably smaller than those from Maurer & Pontil (2009), as illustrated in Section 5. Furthermore, it is unclear how to derive (anytime-valid) inequalities under martingale dependence using the tools from Maurer & Pontil (2009, Theorem 10) (i.e., self-bounding arguments), but we are able to do this with our proof techniques.

## 5. Experiments

We devote this section to exploring the empirical performance of both the upper and lower confidence intervals presented in this paper. Note that, in order to obtain confidence intervals for the standard deviation using our approach, it suffices to take the square root of

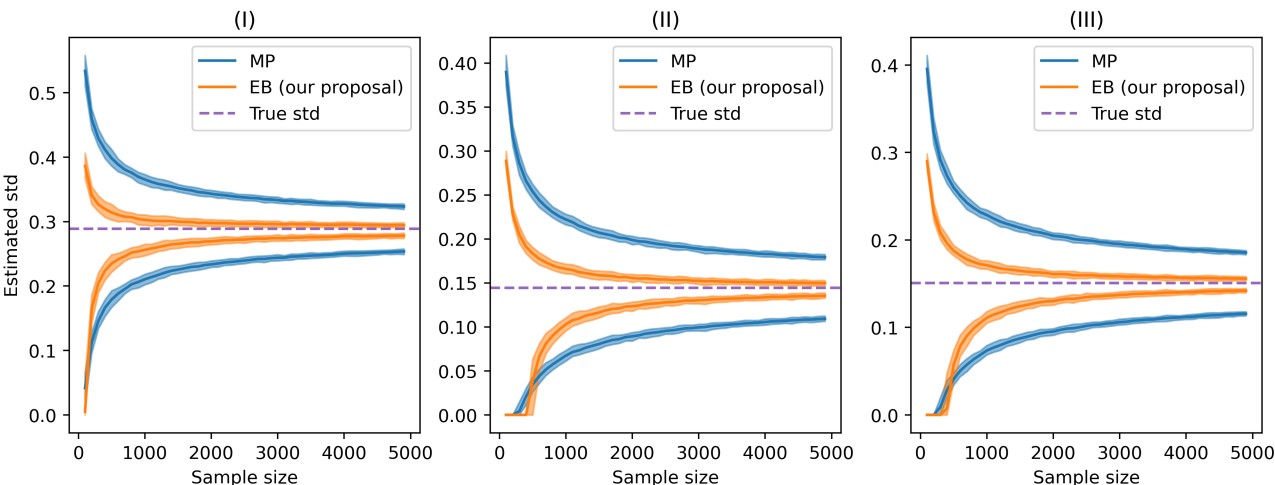

*Figure 1.* Average confidence intervals over 100 simulations for the std $\sigma$ for (I) the uniform distribution in $(0, 1)$, (II) the beta distribution with parameters $(2, 6)$, and (III) the beta distribution with parameters $(5, 5)$. For each of the inequalities, the $0.95\%$-empirical quantiles are also displayed. The Maurer Pontil (MP) inequality (Maurer & Pontil, 2009, Theorem 10) is compared against our proposal (EB). We highlight the improved empirical performance of our methods in all scenarios.

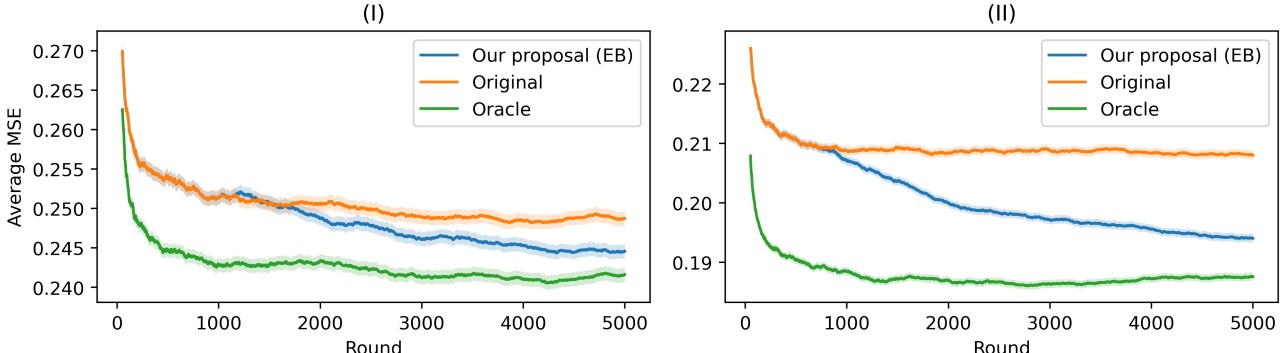

*Figure 2.* Mean squared error of the average treatment effect (ATE) sequential estimations, averaged over $1.5e5$ simulations. The placebo and treated potential outcomes are respectively distributed as (I) a $(0, 0.7)$-uniform distribution and a $(0, 1)$-uniform distribution, and (II) a $(0, 1)$-uniform distribution and a $(2, 6)$-beta distribution. The $0.95\%$-empirical quantiles are also displayed. The original algorithm is compared to a variation that replaces their variance confidence sequences with those from Corollary 4.3, and an oracle algorithm that has access to the variances is also displayed.

the upper and lower confidence intervals for the variance presented in Section 4. In all the experiments, we take $\alpha = 0.05$, and the constants $c_1 = \frac{1}{2}$, $c_2 = \frac{1}{2^4}$, $c_3 = \frac{1}{2^2}$, $c_4 = \frac{1}{2}$, $c_5 = 2$. The code can be found at https://github.com/DMartinezT/emp_bernstein_variance.

**Batch setting.** We compare our results with the inequalities from Maurer & Pontil (2009, Theorem 10), which currently constitute the state of the art for the standard deviation, for fixed sample sizes (the more classical batch setting). Figure 1 displays the average upper and lower confidence intervals for the standard deviation for different samples sizes for three different bounded distributions. Our inequalities consistently demonstrate improved empirical performance in all evaluated scenarios. We defer a comparison with other

alternatives, such as a double empirical Bernstein inequality on the first and second moment, to Appendix F.

**Sequential setting.** We explore a direct algorithmic application of our inequalities in the context of optimal adaptive Neyman allocation for randomized control trials (gold standard in causal inference and off-policy evaluation), following the methodology established in Neopane et al. (2025). In Neopane et al. (2025), the probability of treatment assignment at a given time is built given the confidence sequences for the standard deviations of the treatment and placebo outcomes. In particular, the treatment policy is sequentially updated, and it requires anytime-valid inference on the variance to prove its optimality. We conduct experiments using the confidence sequences used in Neopane et al. (2025),

as well as replacing them with our proposed inequalities, for different potential outcome distributions. We illustrate the results in Figure 2. Note that our inequalities lead to a substantial improved performance of the adaptive Neyman allocation procedure. Especially in later rounds, our inequalities can lead to performances that are closer to those from an oracle algorithm (that knows the variance) than to those from the original algorithm.

## 6. Conclusion

We have provided novel concentration inequalities for the variance of bounded random variables under mild assumptions, instantiating them for both the batch and sequential settings. We have shown their theoretical sharpness, asymptotically matching the first order term of the oracle Bernstein's inequality. Furthermore, our empirical findings demonstrate that they significantly outperform the widely adopted inequalities presented by Maurer & Pontil (2009, Theorem 10).

There are several possible avenues for future work. In Appendix A, we show how the results naturally extend to Hilbert spaces. The proof of Theorem A.3 implicitly exploits the inner product structure of the Hilbert space, which cannot be done in arbitrary Banach spaces. While this challenge may be circumvented by means of the triangle inequality, that approach leads to inflated constants; exploring tighter alternatives for general or smooth Banach spaces would be of interest. Furthermore, the analysis in Section A exploits 'one-dimensional variances'. Extending our work to covariance matrices or operators would also be a natural direction to follow.

## Acknowledgements

DMT thanks Ben Chugg, Ojash Neopane, and Tomás González for insightful conversations. DMT gratefully acknowledges that the project that gave rise to these results received the support of a fellowship from 'la Caixa' Foundation (ID 100010434). The fellowship code is LCF/BQ/EU22/11930075. AR was funded by NSF grant DMS-2310718.

## Impact Statement

This paper presents work whose goal is to advance the field of Machine Learning. There are many potential societal consequences of our work, none which we feel must be specifically highlighted here.

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

## Appendix outline

We organize the appendices as follows. We devote Appendix A to the formalization of the extension of the results to Hilbert spaces. Section B presents auxiliary lemmata that are exploited in the remaining proofs. These are mostly analytic or simple probabilistic results that can be skipped on a first pass. Section C contains more involved technical propositions that are later combined to yield the proofs of Theorem 4.6 and Theorem 4.7. The proofs of such propositions are deferred to Appendix E. Appendix D displays the proofs of the theoretical results exhibited in the main body of the paper, and Appendix F exhibits potential alternative approaches to the proposed empirical Bernstein inequality, illustrating the empirical benefits of the latter.

Throughout, we denote the probability space on which the random variables are defined by $(\Omega, \mathcal{F}, P)$. Furthermore, we use the standard asymptotic big-oh notations. Given functions $f$ and $g$, we write $f(n) = \mathcal{O}(g(n))$ if there exist constants $C, n_0 > 0$ such that $|f(n)| \leq C|g(n)|$ for all $n \geq n_0$. We write $f(n) = \widetilde{\mathcal{O}}(g(n))$ if $f(n) = \mathcal{O}(g(n) \operatorname{polylog}(n))$, where $\operatorname{polylog}(n)$ denotes a polylogarithmic factor in $n$. Finally, we use $f(n) = \Omega(g(n))$ to denote that there exist constants $c, n_0 > 0$ such that $f(n) \geq cg(n)$ for all $n \geq n_0$.

## A. Extension to Hilbert spaces

Our inequalities naturally extend to separable Hilbert spaces. In these more abstract spaces, we shall assume that our random variables lie in a ball of diameter 1, instead of a unit long interval. Similarly, the concept of variance involves norms, instead of just squares of scalars.

**Assumption A.1.** The stream of random variables $X_1, X_2, \ldots$ belongs to a separable Hilbert space $H$, and is such that

$$\|X_t\| \in \left[0, \frac{1}{2}\right], \quad \mathbb{E}_{t-1} X_t = \mu, \quad \mathbb{E}_{t-1} \|X_t - \mu\|^2 = \sigma^2.$$

Under Assumption A.1, all the concentration inequalities for $\sigma^2$ previously presented for the one dimensional case still hold if replacing $(X_i - \bar{\mu}_i)^2$ and $(X_i - \widehat{\mu}_i)^2$ by $\|X_i - \bar{\mu}_i\|^2$ and $\|X_i - \widehat{\mu}_i\|^2$, respectively.

The main technical obstacle of this extension is the generalization of Theorem 3.3 to multivariate settings, which we formalize next. Contrary to its one-dimensional counterpart, its proof builds on more sophisticated techniques from Pinelis (1994); we defer such a proof to Appendix D.7.

**Theorem A.2** (Vector-valued anytime valid Bennett's inequality)**.** *Let $X_1, X_2, \ldots$ be a stream of random variables belonging to a separable Hilbert space $H$ such that $\|X_t\| \leq \frac{1}{2}$, $\mathbb{E}_{t-1} X_t = \mu$, and $\mathbb{E}_{t-1} \|X_t - \mu\|^2 = \sigma^2$, for all $t \geq 1$. For any $\mathbb{R}_+$-valued predictable sequence $(\tilde{\lambda}_i)_{i \geq 1}$, the sequence of sets of $x$ such that*

$$\left\| x - \frac{\sum_{i \leq t} \tilde{\lambda}_i X_i}{\sum_{i \leq t} \tilde{\lambda}_i} \right\| \leq \frac{\log(2/\delta) + \sigma^2 \sum_{i \leq t} \psi_P(\tilde{\lambda}_i)}{\sum_{i \leq t} \tilde{\lambda}_i}$$

*is a $1 - \delta$ confidence sequence for $\mu$.*

The remainder of the one-dimensional results can be extended with relative ease, and so we emphasize once again that the concentration inequalities previously introduced still hold in Hilbert spaces. We highlight that the inequalities are also sharp in this vector-valued setting, with the analysis conducted in Section 4.4 naturally extending to Hilbert spaces.

### A.1. Formalization of auxiliary results

Throughout, let $H$ be a separable Hilbert space and denote

$$B_r(x) = \{y \in H : \|y - r\| \leq r\}.$$

We remind the reader that the theoretical foundation of the results from Section 4 are namely the scalar-valued anytime valid Bennett's inequality (Theorem 3.3) and the supermartingale construction from Theorem 4.1. Theorem A.2 extended the former to the multivariate setting. The remaining foundational piece is the extension of the supermartingale construction from Theorem 4.1, which we present next[3] and whose proof we defer to Appendix D.8.

---

[3]Throughout, we emphasize the fact that the estimator of the mean belongs to the Hilbert space (in contrary to the estimator of the variance, which still belongs to the real line) by means of the notation $\widehat{\mu}_i^{\mathrm{HS}}$.

**Theorem A.3.** *Let Assumption A.1 hold. For any $B_{\frac{1}{2}}(0)$-valued predictable sequence $(\widehat{\mu}_i^{HS})_{i \geq 1}$, define*

$$\tilde{\sigma}_i^2 = \sigma^2 + \left\| \widehat{\mu}_i^{HS} - \mu \right\|^2.$$

*For any $[0,1]$-valued predictable sequence $(\widehat{\sigma}_i)_{i \geq 1}$ and any $[0,1)$-valued predictable sequence $(\lambda_i)_{i \geq 1}$, the processes*

$$S_t^{\pm, HS} = \exp \left\{ \sum_{i \leq t} \lambda_i \left[ \pm \left\| X_i - \widehat{\mu}_i^{HS} \right\|^2 \mp \tilde{\sigma}_i^2 \right] - \psi_E(\lambda_i) \left[ \left\| X_i - \widehat{\mu}_i^{HS} \right\|^2 - \widehat{\sigma}_i^2 \right]^2 \right\}$$

*for $t \geq 1$ and $S_0^{\pm, HS} = 1$, are nonnegative supermartingales.*

This theorem implies that the upper and lower inequalities previously derived for one dimensional processes equally apply to Hilbert spaces. Denoting

$$R_{t,\alpha}^{HS} := \frac{\log(1/\alpha) + \sum_{i \leq t} \psi_E(\lambda_i) \left( \left\| X_i - \widehat{\mu}_i^{HS} \right\|^2 - \widehat{\sigma}_i^2 \right)^2}{\sum_{i \leq t} \lambda_i},$$

$$D_t^{HS} := \frac{\sum_{i \leq t} \left\| \lambda_i X_i - \widehat{\mu}_i^{HS} \right\|^2}{\sum_{i \leq t} \lambda_i}, \quad E_t^{HS} := \frac{\sum_{i \leq t} \lambda_i \left\| \widehat{\mu}_i^{HS} - \mu \right\|^2}{\sum_{i \leq t} \lambda_i},$$

the following corollary is a direct consequence of Theorem A.3, whose proof is analogous to that of Corollary 4.2.

**Corollary A.4.** *Let Assumption 1.1 hold. For any $[0,1)$-valued predictable sequence $(\lambda_i)_{i \geq 1}$, any $B_{\frac{1}{2}}(0)$-valued predictable sequence $(\widehat{\mu}_i^{HS})_{i \geq 1}$, and any $[0,1]$-valued predictable sequence $(\widehat{\sigma}_i)_{i \geq 1}$, the sequence of sets*

$$\left( D_t^{HS} - E_t^{HS} \pm R_{t, \frac{\alpha}{2}}^{HS} \right)$$

*is a $1 - \alpha$ confidence sequence for $\sigma^2$.*

From Corollary A.4, upper and lower inequalities for the variance can be derived analogously to those presented in Section 4. That is, in order to derive an upper inequalities for the variance, it suffices to ignore the term $E_t^{HS}$.

**Corollary A.5** (Vector-valued upper empirical Bernstein for the variance). *Let Assumption 1.1 hold. For any $B_{\frac{1}{2}}(0)$-valued predictable sequence $(\widehat{\mu}_i^{HS})_{i \geq 1}$, any $[0,1]$-valued predictable sequence $(\widehat{\sigma}_i)_{i \geq 1}$, and any $[0,1)$-valued predictable sequence $(\lambda_i)_{i \geq 1}$, it holds that $\left( -\infty, \overline{U}_t^{HS} \right)$ is a $1 - \alpha$ upper confidence sequence for $\sigma^2$, where*

$$U_t^{HS} := D_t^{HS} + R_{t,\alpha}^{HS}.$$

In order to derive $\tilde{R}_{i,\delta}$ such that $\| \widehat{\mu}_i - \mu \| \leq \tilde{R}_{i,\delta}$ for all $i \geq 1$, we propose to use the vector-valued anytime valid Bennett's inequality from Theorem A.2. That is, take

$$\widehat{\mu}_t^{HS} = \frac{\sum_{i=1}^{t-1} \tilde{\lambda}_i X_i}{\sum_{i=1}^{t-1} \tilde{\lambda}_i}, \quad \tilde{R}_{t,\delta} = \frac{\log(2/\delta) + \sigma^2 \sum_{i=1}^{t-1} \psi_P(\tilde{\lambda}_i)}{\sum_{i=1}^{t-1} \tilde{\lambda}_i}. \tag{7}$$

These choices of $\widehat{\mu}_t^{HS}$ and $\tilde{R}_{t,\delta}$ lead to the same exact definition of $\tilde{A}_t$, $\tilde{B}_{t,\delta}$, $\tilde{C}_{t,\delta}$, $A_t$, $B_{t,\delta}$, and $C_{t,\delta}$ from Section 4. The following corollary follows analogously to its one-dimensional counterpart.

**Corollary A.6** (Vector-valued lower empirical Bernstein for the variance). *Let Assumption 1.1 hold. For the $B_{\frac{1}{2}}(0)$-valued predictable sequence $(\widehat{\mu}_i^{HS})_{i \geq 1}$ defined in (2), any $[0,1]$-valued predictable sequence $(\widehat{\sigma}_i^2)_{i \geq 1}$, any $[0,1)$-valued predictable sequence $(\lambda_i)_{i \geq 1}$, and any $[0,\infty)$-valued predictable sequence $(\tilde{\lambda}_i)_{i \geq 1}$, it holds that $\left( L_t^{HS}, \infty \right)$ is a $1 - \alpha$ lower confidence sequence for $\sigma^2$, where $\alpha_1 + \alpha_2 = \alpha$ and*

$$L_t^{HS} := \frac{-B_{t,\alpha_1} + \sqrt{B_{t,\alpha_1}^2 + 4A_t(D_t^{HS} - C_{t,\alpha_1} - R_{t,\alpha_2}^{HS})}}{2A_t}.$$

We propose to take the plug-ins $(\lambda_i)_{i\geq 1}$ and $(\tilde{\lambda}_i)_{i\geq 1}$ analogously to Section 4. These choices require that the definitions of $\widehat{m}_{4,t}^2$ and $\widehat{\sigma}_t^2$ from Section 4 naturally replace the squares by the squares of the norms. Similarly to Section 4, the choice of the split of $\alpha$ into $\alpha_1$ and $\alpha_2$ is also of importance. In practice, we propose to take $\alpha_1$ and $\alpha_2$ analogously to Section 4. The optimality of the results from Section 4.4 for specific choices of $\alpha_1$ and $\alpha_2$ extends analogously to Hilbert spaces. However, Assumption 4.5 ought to be replaced by the following assumption.

**Assumption A.7.** $X_1, X_2, \ldots$ is such that $\mathbb{V}_{i-1}\left[\|X_i - \mu\|^2\right]$ is constant across $i$.

Under Assumption A.1 and Assumption A.7, the first order term of the width of the confidence intervals can be compared with that from the oracle Bernstein-type inequality, i.e.,

$$\sqrt{2\mathbb{V}\left[\|X_i - \mu\|^2\right]\log(1/\alpha)}.$$

The following corollary establishes that the first order width of the confidence intervals does indeed match this oracle benchmark. Its proof is not provided given that it is completely analogous to that of Section 4.4.

**Corollary A.8** (Sharpness). *Let Assumption A.1 and Assumption A.7 hold. If $\alpha = \alpha_{1,n} + \alpha_{2,n}$ as defined in (6), then*

$$\sqrt{n}(U_n^{HS} - D_n^{HS}) \overset{a.s.}{\to} \sqrt{2\mathbb{V}\left[\|X_i - \mu\|^2\right]\log(1/\alpha)},$$
$$\sqrt{n}(D_n^{HS} - L_n^{HS}) \overset{a.s.}{\to} \sqrt{2\mathbb{V}\left[\|X_i - \mu\|^2\right]\log(1/\alpha)}.$$

## B. Auxiliary lemmata

**Lemma B.1.** *For $a \in [0,1]$ and $b \geq 0$,*

$$\psi_P(ab) \leq a^2\psi_P(b), \quad \psi_E(ab) \leq a^2\psi_E(b).$$

*Proof.* It suffices to observe that

$$\psi_P(ab) = \sum_{k=2}^{\infty} \frac{(ab)^k}{k!} \overset{(i)}{\leq} a^2 \sum_{k=2}^{\infty} \frac{b^k}{k!} = a^2\psi_P(b),$$

as well as

$$\psi_E(ab) = \sum_{k=2}^{\infty} \frac{(ab)^k}{k} \overset{(i)}{\leq} a^2 \sum_{k=2}^{\infty} \frac{b^k}{k} = a^2\psi_E(b),$$

where in both (i) follows from $|a| \leq 1$. $\qquad\square$

**Lemma B.2.** *Let $\psi_N = \frac{\lambda^2}{2}$ and $\psi_E(\lambda) = -\log(1-\lambda) - \lambda$. The function $\lambda \in [0,1) \mapsto \frac{\psi_E(\lambda)}{\psi_N(\lambda)}$ is increasing.*

*Proof.* It suffices to observe that

$$\frac{\psi_E(\lambda)}{\psi_N(\lambda)} = \frac{\sum_{k\geq 2}\frac{\lambda^k}{k}}{\frac{\lambda^2}{2}} = 2\sum_{k\geq 2}\frac{\lambda^{k-2}}{k},$$

which is clearly increasing on $\lambda$. $\qquad\square$

**Lemma B.3.** *It holds that*

$$\sum_{i=1}^{n}\frac{1}{\sqrt{i}} \in \left[2\sqrt{n} - 2, 2\sqrt{n} - 1\right],$$

*and so*

$$\frac{1}{\sqrt{n}}\sum_{i=1}^{n}\frac{1}{\sqrt{i}} \to 2.$$

*Proof.* Given that $x \mapsto \frac{1}{\sqrt{x}}$ is a decreasing function, it follows that

$$\int_1^n \frac{1}{\sqrt{x}} dx \leq \sum_{i=1}^n \frac{1}{\sqrt{i}} \leq 1 + \int_1^n \frac{1}{\sqrt{x}} dx,$$

with $\int_1^n \frac{1}{\sqrt{x}} dx = 2\sqrt{n} - 2$. In order to conclude the proof, it suffices to note that

$$\frac{1}{\sqrt{n}} \sum_{i=1}^n \frac{1}{\sqrt{i}} \in \left[ 2 - \frac{2}{\sqrt{n}}, 2 - \frac{1}{\sqrt{n}} \right], \quad 2 - \frac{2}{\sqrt{n}} \to 2, \quad 2 - \frac{1}{\sqrt{n}} \to 2,$$

and invoke the sandwich theorem. $\qquad\square$

**Lemma B.4.** *It holds that*

$$\sum_{i=1}^n \frac{1}{i} \in [\log n, \log n + 1],$$

*and so*

$$\frac{1}{\log n} \sum_{i=1}^n \frac{1}{i} \to 1.$$

*Proof.* Given that $x \mapsto \frac{1}{x}$ is a decreasing function, it follows that

$$\int_1^n \frac{1}{x} dx \leq \sum_{i=1}^n \frac{1}{i} \leq 1 + \int_1^n \frac{1}{x} dx,$$

with $\int_1^n \frac{1}{x} dx = \log n$. In order to conclude the proof, it suffices to note that

$$\frac{1}{\log n} \sum_{i=1}^n \frac{1}{i} \in \left[ 1, 1 + \frac{1}{\log n} \right], \quad 1 + \frac{1}{\log n} \to 1,$$

and invoke the sandwich theorem. $\qquad\square$

**Lemma B.5.** *It holds that*

$$\sum_{i=1}^n \sqrt{i} \in \left[ \frac{1}{3} + \frac{2}{3} n^{\frac{3}{2}}, -\frac{2}{3} + \frac{2}{3}(n+1)^{\frac{3}{2}} \right].$$

*Proof.* Given that $x \mapsto \sqrt{x}$ is an increasing function, it follows that

$$1 + \int_1^n \sqrt{x} dx \leq \sum_{i=1}^n \sqrt{i} \leq \int_1^{n+1} \sqrt{x} dx,$$

with $\int_1^n \sqrt{x} dx = \frac{2}{3} n^{\frac{3}{2}}$. $\qquad\square$

**Lemma B.6.** *It holds that*

$$\sum_{i=2}^\infty \frac{1}{i \log i} = \infty,$$

*and so*

$$\sum_{i=1}^\infty \frac{1}{i \log(i+1)} = \infty.$$

*Proof.* Given that $x \mapsto \frac{1}{x \log x}$ is a decreasing function, it follows that

$$\sum_{i=2}^{n-1} \frac{1}{i \log i} \geq \int_2^n \frac{1}{x \log x} dx \overset{(i)}{=} \int_{\log 2}^{\log n} \frac{1}{u} du,$$

where we have used the change of variable $u = \log x$ in (i). Thus

$$\sum_{i=2}^{\infty} \frac{1}{i \log i} = \lim_{n \to \infty} \sum_{i=2}^{n-1} \frac{1}{i \log i} \geq \lim_{n \to \infty} \int_{\log 2}^{\log n} \frac{1}{u} du = \infty.$$

It remains to note that

$$\sum_{i=1}^{\infty} \frac{1}{i \log(i+1)} \geq \sum_{i=1}^{\infty} \frac{1}{(i+1) \log(i+1)} = \sum_{i=2}^{\infty} \frac{1}{i \log i}.$$

$\square$

**Lemma B.7.** *Let $(a_n)_{n \geq 0}$ be a deterministic sequence such that $a_0 \geq 2$ and $a_n \in [0, 1]$ for $n \geq 1$. Then*

$$\frac{1}{n} \sum_{i=1}^n \frac{a_i}{\sum_{j \leq i-1} a_j} \leq \log \left( \sum_{i=0}^n a_i \right).$$

*Proof.* Denoting $s_i = \sum_{j=0}^i a_j$, it follows that

$$\frac{1}{n} \sum_{i=1}^n \frac{a_i}{\sum_{j \leq i-1} a_j} = \frac{1}{n} \sum_{i=1}^n \frac{s_i - s_{i-1}}{s_{i-1}}.$$

We now note that $\frac{s_i - s_{i-1}}{s_{i-1}}$ is the area of a rectangle with width $s_i - s_{i-1}$ and height $\frac{1}{s_{i-1}}$. Define the function $f(x) := \frac{1}{x-1}$, which is decreasing on $x$ and

$$f(s_i) = \frac{1}{s_i - 1} \overset{(i)}{\geq} \frac{1}{(s_{i-1} + 1) - 1} = \frac{1}{s_{i-1}},$$

where (i) follows from $a_i \in [0, 1]$. Thus

$$\frac{1}{n} \sum_{i=1}^n \frac{s_i - s_{i-1}}{s_{i-1}} \leq \int_{s_0}^{s_n} f(x) dx = \int_{s_0}^{s_n} \frac{1}{x-1} dx = \log(s_n - 1) - \log(a_0 - 1)$$

$$\overset{(i)}{\leq} \log(s_n),$$

where $(i)$ follows from $a_0 \geq 2$, thus concluding the result. $\square$

**Lemma B.8.** *Let $(a_n)_{n \geq 1}$ be a deterministic sequence such that $a_n \to a$. Then*

$$\frac{1}{n} \sum_{i \leq n} a_i \overset{n \to \infty}{\to} a.$$

*Further, if $a_n \to 0$ and $|b_n| < C$, then*

$$\frac{1}{n} \sum_{i \leq n} a_i b_i \overset{n \to \infty}{\to} 0.$$

*Proof.* Let $\epsilon > 0$. We want to show that there exists $M \in \mathbb{N}$ such that

$$\left| a - \frac{1}{n} \sum_{i \leq n} a_i \right| \leq \epsilon.$$

- Given that $a_n \to a$, there exists $M_1 \in \mathbb{N}$ such that $|a_n - a| \leq \frac{\epsilon}{2}$ for all $n \geq M_1$.

- Further, there exists $M_2 \in \mathbb{N}$ such that $\frac{1}{n} \sum_{i=1}^{M_1-1} |a_i - a| \leq \frac{\epsilon}{2}$ for all $n \geq M_2$.

Taking $M = \max\{M_1, M_2\}$, it follows that

$$\left| a - \frac{1}{n} \sum_{i \leq n} a_i \right| \leq \frac{1}{n} \sum_{i \leq n} |a - a_i|$$

$$= \frac{1}{n} \sum_{i=1}^{M_1-1} |a - a_i| + \frac{1}{n} \sum_{i=M_1}^{n} |a - a_i|$$

$$\leq \frac{\epsilon}{2} + \frac{1}{n} \sum_{i=M_1}^{n} \frac{\epsilon}{2} \leq \frac{\epsilon}{2} + \frac{1}{n} \sum_{i=1}^{n} \frac{\epsilon}{2} = \epsilon,$$

thus concluding the first result.

The second result trivially follows after observing

$$\left| \frac{1}{n} \sum_{i \leq n} a_i b_i \right| \leq C \frac{1}{n} \sum_{i \leq n} |a_i|,$$

and the right hand side converges to $0$ in view of the first result. $\qquad \square$

**Lemma B.9.** *Let $(a_n)_{n \geq 1}$ and $(b_n)_{n \geq 1}$ be two deterministic sequences such that*

$$a_n \overset{n \to \infty}{\to} a, \quad b_i \geq 0, \quad \frac{1}{n} \sum_{i=1}^{n} b_i \overset{n \to \infty}{\to} b.$$

*Then,*

$$\frac{1}{n} \sum_{i=1}^{n} a_i b_i \overset{n \to \infty}{\to} ab.$$

*Proof.* Let $\epsilon \in (0, 1)$ be arbitrary. It suffices to show that there exists $M \in \mathbb{N}$ such that

$$\left| \frac{1}{n} \sum_{i=1}^{n} a_i b_i - ab \right| \leq \epsilon$$

for all $n \geq M$. Given that $a_n \to a$, there exists $M_1 \in \mathbb{N}$ such that $\sup_{i \geq M_1} |a_i - a| \leq \frac{\epsilon}{3(b+1)}$. Furthermore, $\frac{1}{n} \sum_{i=1}^{n} b_i \to b$ implies the existence of $M_2 \in \mathbb{N}$ such that

$$\left| \frac{1}{n} \sum_{i=1}^{n} b_i - b \right| \leq \frac{\epsilon}{3(|a| + 1)}.$$

Lastly, there exists $M_3 \in \mathbb{N}$ such that

$$\frac{1}{n} \sup_{i \leq M'-1} |a_i - a| \sum_{i=1}^{M'-1} b_i \leq \frac{\epsilon}{3},$$

where $M' = M_1 \vee M_2$. Taking $M = M' \vee M_3$, it follows that

$$
\begin{aligned}
\left| \frac{1}{n} \sum_{i=1}^{n} a_i b_i - ab \right| &= \left| \frac{1}{n} \sum_{i=1}^{n} a_i b_i - ab_i + ab_i - ab \right| \\
&\leq \frac{1}{n} \sup_{i \leq n} |a_i - a| \sum_{i=1}^{n} b_i + |a| \left| \frac{1}{n} \sum_{i=1}^{n} b_i - b \right| \\
&\leq \frac{1}{n} \sup_{i \leq n} |a_i - a| \sum_{i=1}^{n} b_i + |a| \frac{\epsilon}{3(|a|+1)} \\
&\leq \frac{1}{n} \sup_{i \leq M'-1} |a_i - a| \sum_{i=1}^{M'-1} b_i + \frac{1}{n} \sup_{M' \leq i \leq n} |a_i| \sum_{i=M'}^{n} b_i + \frac{\epsilon}{3} \\
&\leq \frac{\epsilon}{3} + \sup_{i \geq M'} |a_i - a| \frac{1}{n} \sum_{i=1}^{n} b_i + \frac{\epsilon}{3} \\
&\leq \frac{\epsilon}{3} + \frac{\epsilon}{3(b+1)} \left( \frac{\epsilon}{3(|a|+1)} + b \right) + \frac{\epsilon}{3} \leq \frac{\epsilon}{3} + \frac{\epsilon}{3} + \frac{\epsilon}{3} = \epsilon.
\end{aligned}
$$

$\square$

**Lemma B.10.** *Let $(a_{n,i})_{n \geq 1, i \in [n]}$ and $(b_n)_{n \geq 1}$ be two deterministic sequences such that*

$$
a_{n,n} \overset{n \to \infty}{\to} a, \quad |a_{n,i} - a| \leq |a_{i,i} - a|, \quad b_i \geq 0, \quad \frac{1}{n} \sum_{i=1}^{n} b_i \overset{n \to \infty}{\to} b.
$$

*Then*

$$
\frac{1}{n} \sum_{i=1}^{n} a_{n,i} b_i \overset{n \to \infty}{\to} ab.
$$

*Proof.* Let $\epsilon \in (0, 1)$ be arbitrary. It suffices to show that there exists $M \in \mathbb{N}$ such that

$$
\left| \frac{1}{n} \sum_{i=1}^{n} a_{n,i} b_i - ab \right| \leq \epsilon
$$

for all $n \geq M$. Given that $a_{n,n} \to a$, there exists $M_1 \in \mathbb{N}$ such that $\sup_{i \geq M_1} |a_{i,i} - a| \leq \frac{\epsilon}{3(b+1)}$. Furthermore, $\frac{1}{n} \sum_{i=1}^{n} b_i \to b$ implies the existence of $M_2 \in \mathbb{N}$ such that

$$
\left| \frac{1}{n} \sum_{i=1}^{n} b_i - b \right| \leq \frac{\epsilon}{3(|a|+1)}.
$$

Lastly, there exists $M_3 \in \mathbb{N}$ such that

$$
\frac{1}{n} \sup_{i \leq M'-1} |a_{i,i} - a| \sum_{i=1}^{M'-1} b_i \leq \frac{\epsilon}{3},
$$

where $M' = M_1 \vee M_2$. Taking $M = M' \vee M_3$, it follows that

$$
\left| \frac{1}{n} \sum_{i=1}^n a_{n,i} b_i - ab \right| = \left| \frac{1}{n} \sum_{i=1}^n a_{n,i} b_i - ab_i + ab_i - ab \right|
$$

$$
\leq \frac{1}{n} \sup_{i \leq n} |a_{n,i} - a| \sum_{i=1}^n b_i + |a| \left| \frac{1}{n} \sum_{i=1}^n b_i - b \right|
$$

$$
\leq \frac{1}{n} \sup_{i \leq n} |a_{i,i} - a| \sum_{i=1}^n b_i + |a| \frac{\epsilon}{3(|a| + 1)}
$$

$$
\leq \frac{1}{n} \sup_{i \leq M'-1} |a_{i,i} - a| \sum_{i=1}^{M'-1} b_i + \frac{1}{n} \sup_{M' \leq i \leq n} |a_i| \sum_{i=M'}^n b_i + \frac{\epsilon}{3}
$$

$$
\leq \frac{\epsilon}{3} + \sup_{i \geq M'} |a_{i,i} - a| \frac{1}{n} \sum_{i=1}^n b_i + \frac{\epsilon}{3}
$$

$$
\leq \frac{\epsilon}{3} + \frac{\epsilon}{3(b+1)} \left( \frac{\epsilon}{3(|a|+1)} + b \right) + \frac{\epsilon}{3} \leq \frac{\epsilon}{3} + \frac{\epsilon}{3} + \frac{\epsilon}{3} = \epsilon.
$$

$\square$

**Lemma B.11.** *Let $(a_{n,i})_{n \geq 1, i \in [n]}$ such that $a_{n,i} \geq 0$, $\sum_{i=1}^n a_{n,i} \leq C$ for some $C < \infty$,*

$$
a_{n,i} \overset{n \to \infty}{\to} 0 \quad \forall i \geq 1,
$$

*and $(b_n)_{n \geq 1}$ such that $b_n \overset{n \to \infty}{\to} 0$. Then,*

$$
\sum_{i=1}^n a_{n,i} b_i \overset{n \to \infty}{\to} 0.
$$

*Proof.* Let $\epsilon > 0$. We want to show that there exists $M \in \mathbb{N}$ such that $\sum_{i=1}^n a_{n,i} b_i \leq \epsilon$ for all $n \geq M$.

- Given that $b_n \to 0$, there exists $M_1 \in \mathbb{N}$ such that $|b_n| \leq \frac{\epsilon}{2C}$ for all $n > M_1$.

- Further, there exists $M_2 \in \mathbb{N}$ such that $\sum_{i=1}^{M_1} a_{n,i} b_i \leq \frac{\epsilon}{2}$ for all $n \geq M_2$. Such an $M_2$ exists, as it suffices to take $M_2 = \max\{M_{2,i} : i \in [M_1]\}$, where $M_{2,i}$ is such that $a_{n,i} b_i \leq \frac{\epsilon}{2M_1}$ (whose existence is granted by $a_{n,i} \to 0$ as $n \to \infty$ for any fixed $i$).

Taking $M = \max\{M_1, M_2\}$, it follows that

$$
\sum_{i=1}^n a_{n,i} b_i = \sum_{i=1}^{M_1} a_{n,i} b_i + \sum_{i=M_1+1}^n a_{n,i} b_i
$$

$$
\leq \frac{\epsilon}{2} + \frac{\epsilon}{2C} \sum_{i=M_1+1}^n a_{n,i}
$$

$$
\overset{(i)}{\leq} \frac{\epsilon}{2} + \frac{\epsilon}{2C} \sum_{i=1}^n a_{n,i} \leq \frac{\epsilon}{2} + \frac{\epsilon}{2C} C = \epsilon,
$$

where $(i)$ follows from $a_{n,i} \geq 0$, thus concluding the result. $\square$

**Lemma B.12.** *Let $a > 0$ and $b > 0$. If $Z_n > 0$ a.s. and $Z_n \to b$ a.s., then*

$$
\inf_{n \geq 1} \frac{a}{n+1} + \frac{n}{n+1} Z_n
$$

*is strictly positive almost surely.*

*Proof.* Given that $Z_n > 0$ a.s. and $Z_n \to b$ a.s., there exists $A \in \mathcal{F}$ such that $P(A) = 1$ and $Z_n(\omega) > 0$ for all $n$, as well as $Z_n(\omega) \to b$ with $n \to \infty$, for all $\omega \in A$. It suffices to show that, for $\omega \in A$,

$$\inf_{n \geq 1} \frac{a}{n+1} + \frac{n}{n+1} Z_n(\omega) > 0.$$

In order to see this, observe that $Z_n(\omega) \to b$ implies that there exists $m \in \mathbb{N}$ such that $Z_n > \frac{b}{2}$ for all $n \geq m$. Given that the function $x \mapsto x/(x+1)$ is increasing on $n$, for all $n \geq m$,

$$\frac{a}{n+1} + \frac{n}{n+1} Z_n(\omega) \geq \frac{n}{n+1} Z_n(\omega) \geq \frac{m}{m+1} Z_n(\omega) \geq \frac{m}{m+1} \frac{b}{2}.$$

If $Z_n(w) > 0$, then for all $n < m$,

$$\frac{a}{n+1} + \frac{n}{n+1} Z_n(\omega) \geq \frac{a}{n+1} \geq \frac{a}{m}.$$

From these two inequalities, we conclude that

$$\inf_{n \geq 1} \frac{a}{n+1} + \frac{n}{n+1} Z_n(\omega) \geq \frac{a}{m} \wedge \frac{m}{m+1} \frac{b}{2}$$

for all $\omega \in A$. $\qquad \square$

# C. Auxiliary propositions

The proofs of the propositions exhibited herein are deferred to Appendix E. We start by presenting a proof of the almost sure convergence of the fourth moment estimator used throughout. Its proof can be found in Appendix E.1

**Proposition C.1.** *Let $X_1, \ldots, X_n$ fulfill Assumption 1.1 and Assumption 4.5. Let $(\widehat{\mu}_i)_{i \in [n]}$ and $(\widehat{\sigma}_i^2)_{i \in [n]}$ be $[0,1]$-valued predictable sequences. If*

$$\widehat{\mu}_n \overset{a.s.}{\to} \mu, \quad \widehat{\sigma}_n^2 \overset{a.s.}{\to} \sigma^2,$$

*then*

$$\frac{1}{n} \sum_{i=1}^{n} \left[ (X_i - \widehat{\mu}_i)^2 - \widehat{\sigma}_i^2 \right]^2 \overset{a.s.}{\to} \mathbb{V}\left[ (X - \mu)^2 \right],$$

*which implies*

$$\widehat{m}_{4,n}^2 \overset{a.s.}{\to} \mathbb{V}\left[ (X - \mu)^2 \right].$$

If $\mathbb{V}\left[ (X - \mu)^2 \right] = 0$, then the fourth moment estimator does not only converge to $0$ almost surely, but it also does it at a $\tilde{\mathcal{O}}(\frac{1}{t})$ rate. We start by formalizing this result when $(\widehat{m}_{4,i}^2)_{i \in [n]}$ is defined as in Section 4.2. Its proof may be found in Appendix E.2

**Proposition C.2.** *Let $X_1, \ldots, X_n$ fulfill Assumption 1.1 and Assumption 4.5 such that $\mathbb{V}\left[ (X - \mu)^2 \right] = 0$. Let $(\widehat{\mu}_i)_{i \in [n]}$, $(\widehat{\sigma}_i^2)_{i \in [n]}$, and $(\widehat{m}_{4,i}^2)_{i \in [n]}$ be defined as in Section 4.2. Then*

$$\widehat{m}_{4,t}^2 = \tilde{\mathcal{O}}\left( \frac{1}{t} \right)$$

*almost surely.*

The result also extends to the estimator $(\widehat{m}_{4,i}^2)_{i \in [n]}$ defined in Section 4.3. We present such an extension next, whose proof we defer to Appendix E.3.

**Proposition C.3.** *Let* $X_1, \ldots, X_n$ *fulfill Assumption 1.1 and Assumption 4.5 such that* $\mathbb{V}\left[(X - \mu)^2\right] = 0$. *Let* $(\widehat{\mu}_i)_{i \in [n]}$, $(\widehat{\sigma}_i^2)_{i \in [n]}$, *and* $(\widehat{m}_{4,i}^2)_{i \in [n]}$ *be defined as in Section 4.3. If* $\log(1/\alpha_{1,n}) = \tilde{\mathcal{O}}(1)$ *and* $0 < \alpha_{1,n} \leq \alpha$, *then*

$$\widehat{m}_{4,t}^2 = \tilde{\mathcal{O}}\left(\frac{1}{t}\right)$$

*almost surely.*

If $\mathbb{V}\left[(X - \mu)^2\right] = 0$, the (normalized) sum of the plug-ins also converges almost surely to a tractable quantity. We present this result next, and defer the proof to Appendix E.4.

**Proposition C.4.** *Let* $X_1, \ldots, X_n$ *fulfill Assumption 1.1 and Assumption 4.5 such that* $\mathbb{V}\left[(X_i - \mu)^2\right] > 0$. *Let* $(\delta_n)_{n \geq 1}$ *be a deterministic sequence such that*

$$\delta_n \to \delta > 0, \quad \delta_n > 0.$$

*Define*

$$\lambda_{t,\delta_n} := \sqrt{\frac{2 \log(1/\delta_n)}{\widehat{m}_{4,t}^2 n}} \wedge c_1,$$

*with* $c_1 \in (0, 1]$ *and* $\widehat{m}_{4,t}^2$ *defined as in Section 4 with* $c_2 > 0$. *Then*

$$\frac{1}{\sqrt{n}} \sum_{i=1}^n \lambda_{i,\delta_n} \overset{a.s.}{\to} \sqrt{\frac{2 \log(1/\delta)}{\mathbb{V}\left[(X_i - \mu)^2\right]}}.$$

If $\mathbb{V}\left[(X - \mu)^2\right] > 0$, we study the inverse of the (normalized) sum of the plug-ins. In the next proposition, we prove that such a quantity converges almost surely to 0 at a $\tilde{\mathcal{O}}(\frac{1}{\sqrt{n}})$ rate. Its proof may be found in Appendix E.6.

**Proposition C.5.** *Let* $X_1, \ldots, X_n$ *fulfill Assumption 1.1 and Assumption 4.5 such that* $\mathbb{V}\left[(X_i - \mu)^2\right] = 0$. *Let* $(\delta_n)_{n \geq 1}$ *be a deterministic sequence such that*

$$\delta_n \to \delta > 0, \quad \delta_n > 0.$$

*Define*

$$\lambda_{t,\delta_n} := \sqrt{\frac{2 \log(1/\delta_n)}{\widehat{m}_{4,t}^2 n}} \wedge c_1,$$

*with* $c_1 \in (0, 1)$. *Then*

$$\frac{1}{\frac{1}{\sqrt{n}} \sum_{i=1}^n \lambda_{i,\delta_n}} = \tilde{\mathcal{O}}\left(\frac{1}{\sqrt{n}}\right)$$

*almost surely.*

We now analyze the almost sure converge of the sum of the product of two sequences of random variables, one involving the plug-ins through the function $\psi_E$. Its proof may be found in Appendix E.5

**Proposition C.6.** *Let* $X_1, \ldots, X_n$ *fulfill Assumption 1.1 and Assumption 4.5 such that* $\mathbb{V}\left[(X_i - \mu)^2\right] > 0$. *Let* $(\delta_n)_{n \geq 1}$ *be a deterministic sequence such that*

$$\delta_n \nearrow \delta > 0, \quad \delta_n > 0.$$

*Define*

$$\lambda_{t,\delta_n} := \sqrt{\frac{2 \log(1/\delta_n)}{\widehat{m}_{4,t}^2 n}} \wedge c_1,$$

*with $c_1 > 0$, and $\widehat{m}_{4,t}^2$ defined as in Section 4 with $c_2 > 0$. Let $(Z_n)_{n \geq 1}$ be such that*

$$Z_i \geq 0, \quad \frac{1}{n} \sum_{i=1}^n Z_i \overset{a.s.}{\to} a,$$

*with $a \in \mathbb{R}$. Then*

$$\sum_{i=1}^n \psi_E\left(\lambda_{i,\delta_n}\right) Z_i \overset{a.s.}{\to} \frac{a \log(1/\delta)}{V\left[(X_i - \mu)^2\right]}.$$

Lastly, we present a technical proposition that will be used in the proof of Theorem 4.7. We defer its proof to Appendix E.7.

**Proposition C.7.** *Let $\alpha_{1,n} = \Omega\left(\frac{1}{\log(n)}\right)$ and $\widehat{\sigma}_k$ be defined as in Section 4, with $c_3 > 0$. Then*

$$\sum_{2 \leq i \leq n} \frac{\left\{\log(2/\alpha_{1,n}) + \sigma^2 \sum_{k=1}^{i-1} \psi_P\left(\sqrt{\frac{2\log(2/\alpha_{1,n})}{\widehat{\sigma}_k^2 k \log(1+k)}} \wedge c_5\right)\right\}^2}{\left(\sum_{k=1}^{i-1} \sqrt{\frac{2\log(2/\alpha_{1,n})}{\widehat{\sigma}_k^2 k \log(1+k)}} \wedge c_5\right)^2} = \tilde{\mathcal{O}}\left(1\right) \tag{8}$$

*almost surely.*

## D. Main proofs

### D.1. Proof of Theorem 3.3

Fix $t$ and observe

$$
\begin{aligned}
\mathbb{E}_{t-1} \exp\left(\tilde{\lambda}_t(X_t - \mu)\right) &= \mathbb{E}_{t-1} \sum_{k=0}^{\infty} \frac{\left(\tilde{\lambda}_t(X_t - \mu)\right)^k}{k!} \\
&= \sum_{k=0}^{\infty} \frac{\tilde{\lambda}_t^k \mathbb{E}_{t-1}\left[(X_t - \mu)^k\right]}{k!} \\
&\overset{(i)}{=} 1 + \sum_{k=2}^{\infty} \frac{\tilde{\lambda}_t^k \mathbb{E}_{t-1}\left[(X_t - \mu)^k\right]}{k!} \\
&\overset{(ii)}{\leq} 1 + \mathbb{E}_{t-1}\left[(X_t - \mu)^2\right] \sum_{k=2}^{\infty} \frac{\tilde{\lambda}_t^k}{k!} \\
&\overset{(iii)}{=} 1 + \sigma^2 \psi_P(\tilde{\lambda}_t) \\
&\overset{(iv)}{\leq} \exp\left(\sigma^2 \psi_P(\tilde{\lambda}_t)\right),
\end{aligned}
$$

where (i) follows from $\mathbb{E}X_t = \mu$, (ii) from $|X_t - \mu| \leq 1$, (iii) from $\mathbb{E}_{t-1}\left[(X_t - \mu)^2\right] = \sigma^2$, and (iv) from $1 + x \leq \exp(x)$ for all $x \in \mathbb{R}$.

It thus follows that

$$S_t' = \exp\left(\sum_{i \leq t} \tilde{\lambda}_i(X_i - \mu) - \sigma^2 \sum_{i \leq t} \psi_P(\tilde{\lambda}_i)\right) \quad t \geq 1, \quad S_0' = 1,$$

is a nonnegative supermartingale. In view of Ville's inequality (Theorem 3.1), we observe that

$$\mathbb{P}\left(\exp\left\{\sum_{i \leq t} \tilde{\lambda}_i(X_i - \mu) - \sigma^2 \sum_{i \leq t} \psi_P(\tilde{\lambda}_i)\right\} \geq 2/\delta\right) \leq \frac{\delta}{2},$$

thus

$$\mathbb{P}\left(\sum_{i\leq t}\tilde{\lambda}_i(X_i-\mu)-\sigma^2\sum_{i\leq t}\psi_P(\tilde{\lambda}_i)\geq\log(2/\delta)\right)\leq\frac{\delta}{2},$$

and so

$$\mathbb{P}\left(\mu\leq\frac{\sum_{i\leq t}\tilde{\lambda}_iX_i-\sigma^2\sum_{i\leq t}\psi_P(\tilde{\lambda}_i)-\log(2/\delta)}{\sum_{i\leq t}\tilde{\lambda}_i}\right)\leq\frac{\delta}{2}.$$

Arguing analogously replacing $X_i-\mu$ for $\mu-X_i$ and taking the union bound concludes the proof.

### D.2. Proof of Theorem 4.1

The processes are clearly nonnegative, so it remains to prove that they are supermartingales. Let us begin by showing that $S_t^+$ is indeed a supermartingale, i.e.,

$$\mathbb{E}_{t-1}\exp\left\{\lambda_t\left[(X_t-\widehat{\mu}_t)^2-\tilde{\sigma}_t^2\right]-\psi_E(\lambda_t)\left((X_t-\widehat{\mu}_t)^2-\widehat{\sigma}_t^2\right)^2\right\}\leq1 \tag{9}$$

for any $t\geq1$. In order to see this, denote

$$Y_t=(X_t-\widehat{\mu}_t)^2-\tilde{\sigma}_t^2,\quad\delta_t=\widehat{\sigma}_t^2-\tilde{\sigma}_t^2,$$

and restate (9) as

$$\mathbb{E}_{t-1}\exp\left\{\lambda_tY_t-\psi_E(\lambda_t)\left(Y_t-\delta_t\right)^2\right\}\leq1. \tag{10}$$

From Fan et al. (2015, Proposition 4.1), which establishes that $\exp\left\{\xi\lambda-\xi^2\psi_E(\lambda)\right\}\leq1+\xi\lambda$ for any $\lambda\in[0,1)$ and $\xi\geq-1$, it follows that

$$\begin{aligned}
&\mathbb{E}_{t-1}\exp\left\{\lambda_tY_t-\psi_E(\lambda_t)\left(Y_t-\delta_t\right)^2\right\}\\
&=\exp(\lambda_t\delta_t)\mathbb{E}_{t-1}\exp\left\{\lambda_t(Y_t-\delta_t)-\psi_E(\lambda_t)\left(Y_t-\delta_t\right)^2\right\}\\
&\leq\exp(\lambda_t\delta_t)\mathbb{E}_{t-1}\left[1+\lambda_t(Y_t-\delta_t)\right]\\
&\overset{(i)}{=}\exp(\lambda_t\delta_t)\left(1-\lambda_t\delta_t\right)\\
&\overset{(ii)}{\leq}\exp(\lambda_t\delta_t)\exp\left(-\lambda_t\delta_t\right)\\
&=1,
\end{aligned}$$

where (i) is obtained given that $\mathbb{E}_{t-1}Y_t=0$, and (ii) from $1+x\leq e^x$ for all $x\in\mathbb{R}$.

Showing that $S_t^-$ is a supermartingale follows analogously, but replacing $Y_t$ and $\delta_t$ by

$$-(X_t-\widehat{\mu}_t)^2+\tilde{\sigma}_t^2,\quad-\widehat{\sigma}_t^2+\tilde{\sigma}_t^2.$$

Note that this proof is analogous to the proof of Waudby-Smith & Ramdas (2024, Theorem 2), but with non-constant conditional expectations $\tilde{\sigma}_i^2$.

### D.3. Proof of Corollary 4.2

In view of Ville's inequality (Theorem 3.1) and Theorem 4.1, the probability of the event

$$\exp\left\{\sum_{i\leq t}\lambda_i\left[\pm(X_i-\widehat{\mu}_i)^2\mp\tilde{\sigma}_i^2\right]-\psi_E(\lambda_i)\left[(X_i-\widehat{\mu}_i)^2-\widehat{\sigma}_i^2\right]^2\right\}\geq2/\delta$$

uniformly over $t$ is upper bounded by $\frac{\delta}{2}$, and so is

$$\sum_{i \leq t} \lambda_i \left[ \pm (X_i - \widehat{\mu}_i)^2 \mp \widetilde{\sigma}_i^2 \right] - \psi_E(\lambda_i) \left[ (X_i - \widehat{\mu}_i)^2 - \widehat{\sigma}_i^2 \right]^2 \geq \log(2/\delta)$$

uniformly over $t$. Thus

$$\mathbb{P} \left( \sup_t \mp \frac{\sum_{i \leq t} \lambda_i \widetilde{\sigma}_i^2}{\sum_{i \leq t} \lambda_i} \pm D_t - R_{t, \frac{\alpha}{2}} \geq 0 \right) \leq \frac{\delta}{2}.$$

From

$$\frac{\sum_{i \leq t} \lambda_i \widetilde{\sigma}_i^2}{\sum_{i \leq t} \lambda_i} = \frac{\sum_{i \leq t} \lambda_i \left[ \sigma^2 + (\widehat{\mu}_i - \mu)^2 \right]}{\sum_{i \leq t} \lambda_i} = \sigma^2 + E_t,$$

it follows that

$$\mathbb{P} \left( \sup_t \mp \sigma^2 \mp E_t \pm D_t - R_{t, \frac{\alpha}{2}} \geq 0 \right) \leq \frac{\delta}{2},$$

which allows to conclude that

$$\mathbb{P} \left( \sigma \notin \left( D_t - E_t - R_{t, \frac{\alpha}{2}}, D_t - E_t + R_{t, \frac{\alpha}{2}} \right) \right) \leq \frac{\delta}{2},$$

uniformly over $t$.

## D.4. Proof of Corollary 4.4

As exhibited in Section 4.3,

$$\sigma^2 \geq D_t - \frac{\sum_{i \leq t} \lambda_i \widetilde{R}_{i, \alpha_1}^2}{\sum_{i \leq t} \lambda_i} - R_{t, \alpha_2}$$

uniformly over $t$ with probability $\alpha_1 + \alpha_2 = \alpha$. Taking $\widetilde{R}_{i, \alpha_1}^2$ as in (2) leads to

$$\sigma^2 \geq D_t - \frac{\sum_{i \leq t} \lambda_i \left( \widetilde{C}_t^{(2)} \sigma^4 + \widetilde{C}_{t, \alpha_1}^{(1)} \sigma^2 + \widetilde{C}_{t, \alpha_1}^{(0)} \right)}{\sum_{i \leq t} \lambda_i} - R_{t, \alpha_2},$$

i.e.,

$$\sigma^2 \geq D_t - C_t^{(2)} \sigma^4 - \left( C_{t, \alpha_1}^{(1)} - 1 \right) \sigma^2 - C_{t, \alpha_1}^{(0)} - R_{t, \alpha_2}.$$

Thus, it suffices to consider $\sigma^2 \geq \sigma_{l,t}^2$, where $\sigma_{l,t}^2$ is such that

$$\sigma_{l,t}^2 = D_t - C_t^{(2)} \sigma_{l,t}^4 - \left( C_{t, \alpha_1}^{(1)} - 1 \right) \sigma_{l,t}^2 - C_{t, \alpha_1}^{(0)} - R_{t, \alpha_2}.$$

Clearly, solving for this quadratic polynomial leads to (3).

## D.5. Proof of Theorem 4.6

We proceed differently for the cases $\mathbb{V} \left[ (X_i - \mu)^2 \right] = 0$ and $\mathbb{V} \left[ (X_i - \mu)^2 \right] > 0$.

**Case 1:** $\mathbb{V} \left[ (X_i - \mu)^2 \right] = 0$. Note that

$$\sqrt{n} R_{n, \delta_n} = \frac{\log(1/\delta_n) + \sum_{i \leq n} \psi_E(\lambda_{i, \delta_n}) \left( (X_i - \widehat{\mu}_i)^2 - \widehat{\sigma}_i^2 \right)^2}{\frac{1}{\sqrt{n}} \sum_{i \leq n} \lambda_{i, \delta_n}}.$$

Denote

$$\nu_i^2 := \left( (X_i - \widehat{\mu}_i)^2 - \widehat{\sigma}_i^2 \right)^2.$$

In view of Proposition C.2 or Proposition C.3, it follows that

$$n\widehat{m}_{4,n}^2 = \widetilde{\mathcal{O}}(1)$$

almost surely, and so there exists $A \in \mathcal{F}$ with $P(A) = 1$ such that $n\widehat{m}_{4,n}^2(\omega) = \widetilde{\mathcal{O}}(1)$ for all $\omega \in A$. For $\omega \in A$, it may be that

$$\sum_{i=1}^{\infty} \nu_i^2(\omega) =: M < \infty \tag{11}$$

or

$$\sum_{i=1}^{\infty} \nu_i^2(\omega) = \infty. \tag{12}$$

If (11) holds, then

$$\sum_{i \leq n} \psi_E(\lambda_{i,\delta_n}(\omega)) \left( (X_i(\omega) - \widehat{\mu}_i(\omega))^2 - \widehat{\sigma}_i^2(\omega) \right)^2 = \sum_{i \leq n} \psi_E(\lambda_{i,\delta_n}(\omega)) \nu_i^2(\omega)$$

$$\leq \psi_E(c_1) \sum_{i \leq n} \nu_i^2(\omega)$$

$$\leq \psi_E(c_1) M,$$

and so $\log(1/\delta_n) + \sum_{i \leq n} \psi_E(\lambda_{i,\delta_n}(\omega)) \left( (X_i(\omega) - \widehat{\mu}_i(\omega))^2 - \widehat{\sigma}_i^2(\omega) \right)^2$ is upper bounded by $\log(1/l) + \psi_E(c_1) M$, where $l = \inf_{n \in \mathbb{N}} \delta_n$ (which is strictly positive given that $\delta_n \to \delta > 0$ and $\delta_n > 0$). If (12) holds, then there exists $m(\omega) \in \mathbb{N}$ such that, for $t \geq m(\omega)$,

$$\widehat{m}_{4,t}^2(\omega) t = c_2 + \sum_{i=1}^{t-1} \nu_i^2(\omega) \geq \frac{2\log(1/l)}{c_1^2}.$$

Thus

$$\sqrt{\frac{2\log(1/\delta_n)}{\widehat{m}_{4,t}^2(\omega) n}} \leq \sqrt{\frac{2\log(1/l)}{\widehat{m}_{4,t}^2(\omega) t}} \leq c_1,$$

and so

$$\lambda_{i,\delta_n}(\omega) = \sqrt{\frac{2\log(1/\delta_n)}{\widehat{m}_{4,t}^2(\omega) n}}$$

for $i \geq m(\omega)$. Denote

$$(I_n(\omega)) := \log(1/\delta_n) + \sum_{i < m(\omega)} \psi_E(\lambda_{i,\delta_n}(\omega)) \left( (X_i(\omega) - \widehat{\mu}_i(\omega))^2 - \widehat{\sigma}_i^2(\omega) \right)^2,$$

$$(II_n(\omega)) := \sum_{i=m(\omega)}^{n} \psi_E(\lambda_{i,\delta_n}) \left( (X_i(\omega) - \widehat{\mu}_i(\omega))^2 - \widehat{\sigma}_i^2(\omega) \right)^2.$$

We observe that

$$(I_n(\omega)) \leq \log(1/l) + \psi_E(c_1) \sum_{i < m(\omega)} \left( (X_i(\omega) - \widehat{\mu}_i(\omega))^2 - \widehat{\sigma}_i^2(\omega) \right)^2,$$

and so it is bounded. Furthermore,

$$
\begin{aligned}
(II_n(\omega)) &= \sum_{i=m(\omega)}^{n} \psi_E\left(\sqrt{\frac{2\log(1/\delta_n)}{\widehat{m}_{4,i}^2(\omega)n}}\right) \left((X_i(\omega) - \widehat{\mu}_i(\omega))^2 - \widehat{\sigma}_i^2(\omega)\right)^2 \\
&\overset{(i)}{\leq} \frac{2\log(1/\delta_n)\psi_E(c_1)}{c_1^2} \sum_{i=m(\omega)}^{n} \frac{1}{\widehat{m}_{4,i}^2(\omega)n} \left((X_i(\omega) - \widehat{\mu}_i(\omega))^2 - \widehat{\sigma}_i^2(\omega)\right)^2 \\
&\leq \frac{2\log(1/l)\psi_E(c_1)}{c_1^2} \sum_{i=m(\omega)}^{n} \frac{1}{\widehat{m}_{4,i}^2(\omega)n} \left((X_i(\omega) - \widehat{\mu}_i(\omega))^2 - \widehat{\sigma}_i^2(\omega)\right)^2 \\
&\leq \frac{2\log(1/l)\psi_E(c_1)}{c_1^2} \sum_{i=m(\omega)}^{n} \frac{1}{i\widehat{m}_{4,i}^2(\omega)} \left((X_i(\omega) - \widehat{\mu}_i(\omega))^2 - \widehat{\sigma}_i^2(\omega)\right)^2 \\
&= \frac{2\log(1/l)\psi_E(c_1)}{c_1^2} \sum_{i=m(\omega)}^{n} \frac{1}{c_2 + \sum_{i=1}^{i-1} \nu_i^2(\omega)} \nu_i^2(\omega) \\
&\overset{(ii)}{\leq} \frac{2\log(1/l)\psi_E(c_1)}{c_1^2} \log\left(c_2 + \sum_{i=1}^{n} \nu_i^2(\omega)\right) \\
&= \frac{2\log(1/l)\psi_E(c_1)}{c_1^2} \log\left(\widehat{m}_{4,n}^2(\omega)n\right)
\end{aligned}
$$

where (i) follows from Lemma B.1 and $c_1 \in (0,1)$, and (ii) follows from Lemma B.7. Given that $m_{4,n}^2(\omega)n = \tilde{\mathcal{O}}(1)$, it follows from the previous inequalities that $(I_n(\omega)) + (II_n(\omega))$ is also $\tilde{\mathcal{O}}(1)$. Consequently, we have shown that regardless of (11) or (12) holding, it follows that

$$
\sum_{i \leq n} \psi_E(\lambda_{i,\delta_n}(\omega)) \left((X_i(\omega) - \widehat{\mu}_i(\omega))^2 - \widehat{\sigma}_i^2(\omega)\right)^2 = \tilde{\mathcal{O}}(1)
$$

for all $\omega \in A$, with $P(A) = 1$. That is,

$$
\log(1/\delta_n) + \sum_{i \leq n} \psi_E(\lambda_{i,\delta_n}) \left((X_i - \widehat{\mu}_i)^2 - \widehat{\sigma}_i^2\right)^2 = \tilde{\mathcal{O}}(1)
$$

almost surely. Further, by Proposition C.5 and $\delta_n \to \delta$, it also follows that

$$
\frac{1}{\frac{1}{\sqrt{n}}\sum_{i=1}^{n}\lambda_{i,\delta_n}} = \tilde{\mathcal{O}}\left(\frac{1}{\sqrt{n}}\right)
$$

almost surely. Thus, it is concluded that

$$
\sqrt{n}R_{n,\delta_n} = \tilde{\mathcal{O}}\left(\frac{1}{\sqrt{n}}\right)
$$

almost surely, and so it converges to 0 almost surely.

**Case 2:** $\mathbb{V}\left[(X_i - \mu)^2\right] > 0$. By Proposition C.4,

$$
\frac{1}{\sqrt{n}}\sum_{i=1}^{n}\lambda_{i,\delta_n} \overset{a.s.}{\to} \sqrt{\frac{2\log(1/\delta)}{\mathbb{V}\left[(X_i - \mu)^2\right]}}.
$$

In view of

$$
\frac{1}{n}\sum_{i \leq n}\left((X_i - \widehat{\mu}_i)^2 - \widehat{\sigma}_i^2\right)^2 \to \mathbb{V}\left[(X_i - \mu)^2\right]
$$

almost surely (Proposition C.1) and Proposition C.6, it follows that

$$\sum_{i \leq n} \psi_E(\lambda_i) \left( (X_i - \widehat{\mu}_i)^2 - \widehat{\sigma}_i^2 \right)^2 \to \frac{V\left[(X_i - \mu)^2\right] \log(1/\delta)}{V\left[(X_i - \mu)^2\right]} = \log(1/\delta).$$

We thus conclude that

$$\sqrt{n} R_{n,\delta_n} \to \frac{\log(1/\delta) + \log(1/\delta)}{\sqrt{\frac{2 \log(1/\delta)}{\mathbb{V}[(X_i - \mu)^2]}}} = \sqrt{2 \mathbb{V}\left[(X_i - \mu)^2\right] \log(1/\delta)}$$

almost surely.

### D.6. Proof of Theorem 4.7

We differentiate the cases $\mathbb{V}\left[(X_i - \mu)^2\right] = 0$ and $\mathbb{V}\left[(X_i - \mu)^2\right] > 0$.

**Case 1:** $\mathbb{V}\left[(X_i - \mu)^2\right] = 0$. By Proposition C.5 and $\alpha_{2,n} \to \alpha$,

$$\frac{1}{\frac{1}{\sqrt{n}} \sum_{i=1}^n \lambda_{i,\alpha_{2,n}}} = \tilde{\mathcal{O}}\left(\frac{1}{\sqrt{n}}\right)$$

almost surely. Furthermore, in view of Proposition C.7,

$$\sum_{i \leq n} \frac{\left\{ \log(2/\alpha_{1,n}) + \sigma^2 \sum_{k=1}^{i-1} \psi_P\left(\sqrt{\frac{2 \log(2/\alpha_{1,n})}{\widehat{\sigma}_k^2 k \log(1+k)}} \wedge c_5\right) \right\}^2}{\left(\sum_{k=1}^{i-1} \sqrt{\frac{2 \log(2/\alpha_{1,n})}{\widehat{\sigma}_k^2 k \log(1+k)}} \wedge c_5\right)^2}.$$

is $\tilde{\mathcal{O}}(1)$ almost surely. Thus, the product of both is $\tilde{\mathcal{O}}\left(\frac{1}{\sqrt{n}}\right)$ almost surely, which further implies that it converges to $0$ almost surely.

**Case 2:** $\mathbb{V}\left[(X_i - \mu)^2\right] > 0$. By Proposition C.4 and $\alpha_{2,n} \to \alpha$,

$$\frac{1}{\sqrt{n}} \sum_{i \leq n} \lambda_{i,\alpha_{2,n}} \overset{a.s.}{\to} \sqrt{\frac{2 \log(1/\delta)}{\mathbb{V}\left[(X_i - \mu)^2\right]}}.$$

Thus, it suffices to prove

$$\sum_{i \leq n} \lambda_{i,\alpha_{2,n}} R_{i,\alpha_{1,n}}^2 \overset{a.s.}{\to} 0 \tag{13}$$

to conclude the proof. We note that $\lambda_{i,\alpha_{2,n}} R_{i,\alpha_{1,n}}^2$ is equal to

$$\frac{1}{\sqrt{n}} \sum_{i \leq n} \sqrt{n} \lambda_{i,\alpha_{2,n}} \frac{\left\{ \log(2/\alpha_{1,n}) + \sigma^2 \sum_{k=1}^{i-1} \psi_P\left(\sqrt{\frac{2 \log(2/\alpha_{1,n})}{\widehat{\sigma}_k^2 k \log(1+k)}} \wedge c_5\right) \right\}^2}{\left(\sum_{k=1}^{i-1} \sqrt{\frac{2 \log(2/\alpha_{1,n})}{\widehat{\sigma}_k^2 k \log(1+k)}} \wedge c_5\right)^2}.$$

By Proposition C.7,

$$\frac{1}{\sqrt{n}} \sum_{i \leq n} \frac{\left\{ \log(2/\alpha_{1,n}) + \sigma^2 \sum_{k=1}^{i-1} \psi_P\left(\sqrt{\frac{2 \log(2/\alpha_{1,n})}{\widehat{\sigma}_k^2 k \log(1+k)}} \wedge c_5\right) \right\}^2}{\left(\sum_{k=1}^{i-1} \sqrt{\frac{2 \log(2/\alpha_{1,n})}{\widehat{\sigma}_k^2 k \log(1+k)}} \wedge c_5\right)^2}.$$

is $\tilde{\mathcal{O}}(\frac{1}{\sqrt{n}})$ almost surely, and so it suffices to show that

$$\sup_{n \in \mathbb{N}} \sup_{i \leq n} \sqrt{n} \lambda_{i,\alpha_{2,n}}$$

is bounded almost surely in order to conclude the result (in view of Hölder's inequality, (13) will follow). Analogously to the proof of Proposition C.4, there exist $m_\omega \in \mathbb{N}$ and $u(\omega) < \infty$ such that

$$\lambda_{t,\alpha_{2,n}}(\omega) = \sqrt{\frac{2\log(1/\alpha_{2,n})}{\widehat{m}_{4,t}^2(\omega)n}}, \quad \frac{1}{\widehat{m}_{4,n}^2(\omega)} \leq u(\omega),$$

for $n \geq m_\omega$ and $\omega \in A$, with $P(A) = 1$. Given that $\alpha_{2,n} \to \alpha > 0$ and $\alpha_{2,n} > 0$, then $l := \inf_n \alpha_{2,n} > 0$, and so we observe that

$$\sqrt{n}\lambda_{t,\alpha_{2,n}}(\omega) = \sqrt{\frac{2\log(1/\alpha_{2,n})}{\widehat{m}_{4,t}^2(\omega)}} \leq \sqrt{2\log(1/l)}u(\omega)$$

for $n \geq m_\omega$. It follows that

$$\sup_{n\in\mathbb{N}}\sup_{i\leq n}\sqrt{n}\lambda_{i,\alpha_{2,n}}(\omega) \leq \left(\sqrt{2\log(1/l)}u(\omega)\right) \vee \left(\sup_{n<m_\omega}\sup_{i\leq n}\sqrt{n}\lambda_{i,\alpha_{2,n}}(\omega)\right) < \infty,$$

and thus the result is concluded by Hölder's inequality.

### D.7. Proof of Theorem A.2

Denote

$$f_t = \sum_{i\leq t}\tilde{\lambda}_i(X_i - \mu).$$

Pinelis (1994, Theorem 3.2) showed that[4]

$$\mathbb{E}_{t-1}\cosh\left(\|f_t\|\right) \leq \left(1 + \mathbb{E}_{t-1}\psi_P\left(\tilde{\lambda}_t\|X_t - \mu\|\right)\right)\cosh\left(\|f_{t-1}\|\right).$$

Similarly to the proof of Theorem 3.3, it now follows that

$$\begin{aligned}
1 + \mathbb{E}_{t-1}\psi_P\left(\tilde{\lambda}_t\|X_t - \mu\|\right) &= 1 + \mathbb{E}_{t-1}\sum_{k=2}^{\infty}\frac{\left(\tilde{\lambda}_t\|X_t - \mu\|\right)^k}{k!} \\
&= 1 + \sum_{k=2}^{\infty}\frac{\tilde{\lambda}_t^k\mathbb{E}_{t-1}\left[\|X_t - \mu\|^k\right]}{k!} \\
&\overset{(i)}{=} 1 + \sum_{k=2}^{\infty}\frac{\tilde{\lambda}_t^k\mathbb{E}_{t-1}\left[\|X_t - \mu\|^k\right]}{k!} \\
&\overset{(ii)}{\leq} 1 + \mathbb{E}_{t-1}\left[\|X_t - \mu\|^2\right]\sum_{k=2}^{\infty}\frac{\tilde{\lambda}_t^k}{k!} \\
&\overset{(iii)}{=} 1 + \sigma^2\psi_P(\tilde{\lambda}_t) \overset{(iv)}{\leq} \exp\left(\sigma^2\psi_P(\tilde{\lambda}_t)\right),
\end{aligned}$$

where (i) follows from $\mathbb{E}X_t = \mu$, (ii) follows from $\|X_t - \mu\| \leq 1$, (iii) follows from $\mathbb{E}_{t-1}\|X_t - \mu\|^2 = \sigma^2$, and (iv) follows from $1 + x \leq \exp(x)$ for all $x \in \mathbb{R}$. Thus, the process

$$S_t' = \cosh\left(\left\|\sum_{i\leq t}\tilde{\lambda}_i(X_i - \mu)\right\|\right)\exp\left(-\sigma^2\sum_{i\leq t}\psi_P(\tilde{\lambda}_i)\right),$$

---

[4]Pinelis (1994, Theorem 3.2) requires separability (required in Pinelis (1994, Lemma 2.3)). Separability is ultimately needed to ensure the tightness of the probability measure on the (more generally) Banach space.

for $t \geq 1$, and $S_0' = 1$, is a nonnegative supermartingale. In view of Ville's inequality (Theorem 3.1) we observe that

$$\mathbb{P}\left(\cosh\left(\left\|\sum_{i \leq t} \tilde{\lambda}_i(X_i - \mu)\right\|\right) \exp\left(-\sigma^2 \sum_{i \leq t} \psi_P(\tilde{\lambda}_i)\right) \geq \frac{1}{\delta}\right) \leq \delta,$$

and, from $\exp x \leq 2 \cosh x$ for $x \in \mathbb{R}$, it follows that

$$\mathbb{P}\left(\exp\left(\left\|\sum_{i \leq t} \tilde{\lambda}_i(X_i - \mu)\right\| - \sigma^2 \sum_{i \leq t} \psi_P(\tilde{\lambda}_i)\right) \geq \frac{2}{\delta}\right) \leq \delta.$$

Thus

$$\mathbb{P}\left(\left\|\sum_{i \leq t} \tilde{\lambda}_i(X_i - \mu)\right\| - \sigma^2 \sum_{i \leq t} \psi_P(\tilde{\lambda}_i) \geq \log(2/\delta)\right) \leq \frac{\delta}{2},$$

and so

$$\mathbb{P}\left(\left\|\mu - \frac{\sum_{i \leq t} \tilde{\lambda}_i X_i}{\sum_{i \leq t} \tilde{\lambda}_i}\right\| \leq \frac{\sigma^2 \sum_{i \leq t} \psi_P(\tilde{\lambda}_i) + \log(2/\delta)}{\sum_{i \leq t} \tilde{\lambda}_i}\right) \leq \frac{\delta}{2}.$$

### D.8. Proof of Theorem A.3

The proof is analogous to that of Theorem 4.1, replacing $Y_t = (X_t - \widehat{\mu}_t)^2 - \tilde{\sigma}_t^2$ by $Y_t = \|X_t - \widehat{\mu}_t\|^2 - \tilde{\sigma}_t^2$.

## E. Proofs of auxiliary propositions

### E.1. Proof of Proposition C.1

Denote $\tilde{m}_{4,n}^2 := \frac{1}{n}\sum_{i=1}^{n}\left[(X_i - \widehat{\mu}_i)^2 - \widehat{\sigma}_i^2\right]^2$. Then

$$\tilde{m}_{4,n}^2 = \frac{1}{n}\sum_{i=1}^{n}\left[(X_i - \widehat{\mu}_i)^2 - \sigma^2 + \sigma^2 - \widehat{\sigma}_i^2\right]^2$$

$$= \underbrace{\frac{1}{n}\sum_{i=1}^{n}\left[(X_i - \widehat{\mu}_i)^2 - \sigma^2\right]^2}_{(I_n)} - \underbrace{\frac{2}{n}\sum_{i=1}^{n}\left[(X_i - \widehat{\mu}_i)^2 - \sigma^2\right]\left(\sigma^2 - \widehat{\sigma}_i^2\right)}_{(II_n)}$$

$$+ \underbrace{\frac{1}{n}\sum_{i=1}^{n}\left(\sigma^2 - \widehat{\sigma}_i^2\right)^2}_{(III_n)}.$$

It suffices to prove that $(I_n)$ converges to $\mathbb{V}\left[(X - \mu)^2\right]$ almost surely, and $(II_n)$ and $(III_n)$ converge to $0$ almost surely.

- Denoting $\gamma_i = (\mu - \widehat{\mu}_i)^2 + 2(X_i - \mu)(\mu - \widehat{\mu}_i)$, it follows that

$$(I_n) = \frac{1}{n}\sum_{i=1}^{n}\left[(X_i - \mu + \mu - \widehat{\mu}_i)^2 - \sigma^2\right]^2$$

$$= \frac{1}{n}\sum_{i=1}^{n}\left[(X_i - \mu)^2 - \sigma^2 + (\mu - \widehat{\mu}_i)^2 + 2(X_i - \mu)(\mu - \widehat{\mu}_i)\right]^2$$

$$= \frac{1}{n}\sum_{i=1}^{n}\left[(X_i - \mu)^2 - \sigma^2 + \gamma_i\right]^2$$

$$= \frac{1}{n}\sum_{i=1}^{n}\left[(X_i - \mu)^2 - \sigma^2\right]^2 + \frac{2}{n}\sum_{i=1}^{n}\left[(X_i - \mu)^2 - \sigma^2\right]\gamma_i + \frac{1}{n}\sum_{i=1}^{n}\gamma_i^2.$$

The first of these three summands converges to $\mathbb{V}\left[(X - \mu)^2\right]$ by the scalar martingale strong law of large numbers (Hall & Heyde, 2014, Theorem 2.1). Given that $(\widehat{\mu}_i - \mu) \to 0$ almost surely, $\gamma_i \to 0$ almost surely as well. Thus, the latter summands converge to 0 almost surely: the second summand converges to 0 in view of Lemma B.8 and the fact that the $(X_i - \mu)^2 - \sigma^2$ are bounded; the third summand converges to 0 almost surely also in view of Lemma B.8.

- Given that $\left[(X_i - \widehat{\mu}_i)^2 - \sigma^2\right]$ is bounded and $\left(\sigma^2 - \widehat{\sigma}_i^2\right) \to 0$ almost surely, $(II_n)$ converges to 0 almost surely by Lemma B.8.

- Given that $\left(\sigma^2 - \widehat{\sigma}_i^2\right) \to 0$ almost surely, $\left(\sigma^2 - \widehat{\sigma}_i^2\right)^2 \to 0$ almost surely, and so $(III_n)$ converges to 0 almost surely by Lemma B.8.

Thus, $\tilde{m}_{4,n}^2 \to \mathbb{V}\left[(X - \mu)^2\right]$ almost surely. Given that $\widehat{m}_{4,n}^2 = \frac{c_2}{n} + \frac{n-1}{n}\tilde{m}_{4,n-1}^2$, this also implies that $\widehat{m}_{4,n}^2 \to \mathbb{V}\left[(X - \mu)^2\right]$ almost surely.

### E.2. Proof of Proposition C.2

Denote $v_i = (X_i - \widehat{\mu}_i)^2 - \widehat{\sigma}_i^2$, so that

$$m_{4,t}^2 = \frac{c_2 + \sum_{i \leq t-1} v_i^2}{t}.$$

If $\sigma^2 = 0$, then $X_i = \mu$ for all $i$. In that case,

$$\widehat{\mu}_i = \frac{c_4}{i} + \frac{i-1}{i}\mu, \quad \widehat{\sigma}_i^2 = \frac{c_3}{i} + \frac{(c_4 - \mu)^2 \sum_{j \leq i-1} \frac{1}{j^2}}{i}.$$

Note that

$$\widehat{\sigma}_i^2 \leq \frac{c_3}{i} + \frac{(c_4 - \mu)^2 \sum_{j=1}^{\infty} \frac{1}{j^2}}{i} = \frac{c_3 + (c_4 - \mu)^2 \frac{\pi^2}{6}}{i},$$

and so

$$
\begin{aligned}
v_i^2 &= \left[\left(\frac{c_4 - \mu}{i}\right)^2 - \widehat{\sigma}_i^2\right]^2 \\
&\overset{(i)}{\leq} 2\left(\frac{c_4 - \mu}{i}\right)^4 + 2\widehat{\sigma}_i^4 \\
&\leq 2\frac{(c_4 - \mu)^4}{i^2} + 2\widehat{\sigma}_i^4 \\
&\leq \frac{\kappa_1}{i^2},
\end{aligned}
$$

where $\kappa_1 := 2(c_4 - \mu)^4 + 2\left(c_3 + (c_4 - \mu)^2 \frac{\pi^2}{6}\right)^2$, and (i) follows from $(a - b)^2 \leq 2a^2 + 2b^2$. Thus

$$m_{4,t}^2 \leq \frac{c_2 + \sum_{i \leq t-1} \frac{\kappa_1}{i^2}}{t} \leq \frac{c_2 + \sum_{i=1}^{\infty} \frac{\kappa_1}{i^2}}{t} = \frac{c_2 + \frac{\kappa_1 \pi^2}{6}}{t} = \mathcal{O}\left(\frac{1}{t}\right).$$

If $\sigma^2 > 0$, note that

$$
\begin{aligned}
v_i &= (X_i - \widehat{\mu}_i)^2 - \widehat{\sigma}_i^2 \\
&= (X_i - \mu)^2 - \sigma^2 + 2(X_i - \mu)(\mu - \widehat{\mu}_i) + (\mu - \widehat{\mu}_i)^2 + \sigma^2 - \widehat{\sigma}_i^2 \\
&\overset{(i)}{=} 2(X_i - \mu)(\mu - \widehat{\mu}_i) + (\mu - \widehat{\mu}_i)^2 + \sigma^2 - \widehat{\sigma}_i^2,
\end{aligned}
$$

where (i) follows from $(X_i - \mu)^2 = \sigma^2$, and so

$$|v_i| \le 3 |\mu - \widehat{\mu}_i| + |\sigma^2 - \widehat{\sigma}_i^2| .$$

The martingale analogue of Kolmogorov's law of iterated logarithm (Stout, 1970) establishes that

$$\limsup_{i \to \infty} |\widehat{\mu}_i - \mu| \frac{\sqrt{n}}{\sqrt{2\sigma^2 \log\log(n\sigma^2)}} = 1$$

almost surely. That implies that there exists $A \in \mathcal{F}$ such that $P(A) = 1$ and, for all $\omega \in A$,

$$|\widehat{\mu}_i(\omega) - \mu| \le \sqrt{C(\omega)} \frac{\sqrt{2\sigma^2 \log\log(i\sigma^2)}}{\sqrt{i}}$$

for some $C(\omega) < \infty$. Furthermore,

$$
\begin{aligned}
\widehat{\sigma}_i^2 &= \frac{c_3 + \sum_{j \le i-1} (X_j - \bar{\mu}_j)^2}{i} \\
&= \frac{c_3 + \sum_{j \le i-1} (X_j - \mu)^2 + 2(X_j - \mu)(\mu - \bar{\mu}_j) + (\mu - \bar{\mu}_j)^2}{i} \\
&= \frac{c_3 + \sum_{j \le i-1} (X_j - \mu)^2 + 2(X_j - \mu)(\mu - \bar{\mu}_j) + (\mu - \bar{\mu}_j)^2}{i} \\
&\stackrel{(i)}{=} \frac{c_3}{i} + \frac{i-1}{i}\sigma^2 + \frac{\sum_{j \le i-1} 2(X_j - \mu)(\mu - \bar{\mu}_j) + (\mu - \bar{\mu}_j)^2}{i},
\end{aligned}
$$

where (i) follows from $(X_i - \mu)^2 = \sigma^2$, and so

$$\sigma^2 - \widehat{\sigma}_i^2 = \frac{\sigma^2}{i} - \frac{c_3}{i} - \frac{\sum_{j \le i-1} 2(X_j - \mu)(\mu - \bar{\mu}_j) + (\mu - \bar{\mu}_j)^2}{i},$$

which implies

$$
\begin{aligned}
|\sigma^2 - \widehat{\sigma}_i^2| &\le \frac{c_3}{i} + \frac{\sigma^2}{i} + \left| \frac{\sum_{j \le i-1} 2(X_j - \mu)(\mu - \bar{\mu}_j) + (\mu - \bar{\mu}_j)^2}{i} \right| \\
&\le \frac{c_3}{i} + \frac{\sigma^2}{i} + 3\frac{\sum_{j \le i-1} |\mu - \bar{\mu}_j|}{i} \\
&\le \frac{2}{i} + 3\frac{\sum_{j \le i-1} |\mu - \bar{\mu}_j|}{i}.
\end{aligned}
$$

Thus, for $\omega \in A$,

$$
\begin{aligned}
|v_i(\omega)| &\le 3 |\mu - \widehat{\mu}_i(\omega)| + |\sigma^2 - \widehat{\sigma}_i^2(\omega)| \\
&\le 3 |\mu - \widehat{\mu}_i(\omega)| + \frac{2}{i} + 3\frac{\sum_{j \le i-1} |\mu - \bar{\mu}_j(\omega)|}{i} \\
&\le 3\frac{\sqrt{2C(\omega)\sigma^2 \log\log(i\sigma^2)}}{\sqrt{i}} + \frac{2}{i} + 3\frac{\sum_{j \le i-1} \sqrt{C(\omega)}\frac{\sqrt{2\sigma^2 \log\log(j\sigma^2)}}{\sqrt{j}}}{i} \\
&\le 3\frac{\sqrt{2C(\omega)\sigma^2 \log\log(i\sigma^2)}}{\sqrt{i}} + \frac{2}{i} + 3\sqrt{2C(\omega)\sigma^2 \log\log(i\sigma^2)}\frac{\sum_{j \le i-1} \frac{1}{\sqrt{j}}}{i} \\
&\stackrel{(i)}{\le} 3\frac{\sqrt{2C(\omega)\sigma^2 \log\log(i\sigma^2)}}{\sqrt{i}} + \frac{2}{\sqrt{i}} + 6\sqrt{2C(\omega)\sigma^2 \log\log(i\sigma^2)}\frac{1}{\sqrt{i}} \\
&= \left(2 + 9\sqrt{2C(\omega)\sigma^2 \log\log(i\sigma^2)}\right)\frac{1}{\sqrt{i}},
\end{aligned}
$$

where (i) follows from Lemma B.3. Thus,

$$(v_i(\omega))^2 \leq \left(2 + 9\sqrt{2C(\omega)\sigma^2 \log\log(i\sigma^2)}\right)^2 \frac{1}{i}.$$

From here, it follows that, for $\omega \in A$,

$$
\begin{aligned}
m_{4,t}^2(\omega) &= \frac{c_2 + \sum_{i \leq t-1} v_i^2(\omega)}{t} \\
&\leq \frac{c_2 + \sum_{i \leq t-1} \left(2 + 9\sqrt{2C(\omega)\sigma^2 \log\log(i\sigma^2)}\right)^2 \frac{1}{i}}{t} \\
&\leq \frac{c_2 + \left(2 + 9\sqrt{2C(\omega)\sigma^2 \log\log(t\sigma^2)}\right)^2 \sum_{i \leq t-1} \frac{1}{i}}{t} \\
&\overset{(i)}{\leq} \frac{c_2 + \left(2 + 9\sqrt{2C(\omega)\sigma^2 \log\log(t\sigma^2)}\right)^2 (1 + \log t)}{t} \\
&= \tilde{\mathcal{O}}\left(\frac{1}{t}\right),
\end{aligned}
$$

where (i) follows from Lemma B.4. Noting that $P(A) = 1$ concludes the result.

### E.3. Proof of Proposition C.3

The proof follows analogously to that of Proposition C.2 as soon as we show that

- if $\sigma = 0$, then

$$(X_i - \widehat{\mu}_i)^2 = 0;$$

- if $\sigma > 0$, then

$$|\widehat{\mu}_i(\omega) - \mu| = \tilde{\mathcal{O}}\left(\frac{1}{\sqrt{i}}\right)$$

  almost surely.

Let us now prove each of the statements.

If $\sigma = 0$, then

$$\widehat{\mu}_t = \frac{\sum_{i=1}^{t-1} \tilde{\lambda}_i X_i}{\sum_{i=1}^{t-1} \tilde{\lambda}_i} = \frac{\sum_{i=1}^{t-1} \tilde{\lambda}_i \mu}{\sum_{i=1}^{t-1} \tilde{\lambda}_i} = \mu = X_i,$$

and so $(X_i - \widehat{\mu}_i)^2 = 0$, and we are done.

If $\sigma > 0$, define

$$\iota_i = \sqrt{\frac{2}{\widehat{\sigma}_i^2 i \log(i+1)}}.$$

We shall start by studying the growth of $\sum_{i \leq n} \iota_i X_i$. In view of

$$
\begin{aligned}
\sum_{i=1}^{\infty} \mathbb{E}_{i-1}\left[\iota_i^2 (X_i - \mu)^2\right] &= \sigma^2 \sum_{i=1}^{\infty} \iota_i^2 = \sigma^2 \sum_{i=1}^{\infty} \frac{2}{\widehat{\sigma}_i^2 i \log(i+1)} \\
&\geq 2\sigma^2 \sum_{i=1}^{\infty} \frac{1}{i \log(i+1)} \overset{(i)}{=} \infty,
\end{aligned}
$$

where (i) follows from Lemma B.6, as well as $|\iota_i(X_i - \mu)| \leq \iota_i$ with $\iota_i \to 0$ almost surely, we can apply the martingale analogue of Kolmogorov's law of the iterated logarithm (Stout, 1970). Thus, defining

$$S_n^2 := \sum_{i=1}^{n} \mathbb{E}_{i-1}\left[\iota_i^2(X_i - \mu)^2\right] = \sigma^2 \sum_{i=1}^{n} \iota_i^2,$$

it follows that

$$\limsup_{n \to \infty} \frac{\sum_{i \leq n} \iota_i(X_i - \mu)}{\sqrt{2S_n^2 \log\log(S_n^2)}} = 1$$

almost surely. Hence, there exists $A_1 \in \mathcal{F}$ with $P(A_1) = 1$ such that

$$\left| \sum_{i \leq n} \iota_i(\omega)(X_i(\omega) - \mu) \right| \leq C(\omega)\sqrt{2S_n^2 \log\log(S_n^2)}$$

for all $\omega \in A_1$. Given that $\widehat{\sigma}_n \to \sigma$ almost surely, there exists $A_2 \in \mathcal{F}$ with $P(A_2) = 1$ such that $\widehat{\sigma}_n(\omega) \to \sigma$. Hence, for each $\omega \in A_2$, there exists $m(\omega) \in \mathbb{N}$ such that $\widehat{\sigma}_i \geq \frac{\sigma}{2}$ for all $i \geq m(\omega)$. Thus, for $\omega \in A_2$,

$$S_n^2(\omega) = \sigma^2 \sum_{i=1}^{n} \frac{2}{\widehat{\sigma}_i^2(\omega)i \log(i+1)}$$
$$\leq \sum_{i=1}^{n} \frac{8}{i \log(i+1)}$$
$$\leq \sum_{i=1}^{n} \frac{16}{i}$$
$$\overset{(i)}{\leq} 16(\log n + 1),$$

where (i) is obtained in view of Lemma B.4. Hence, for $\omega \in A := A_1 \cap A_2$,

$$\left| \sum_{i \leq n} \iota_i(\omega)(X_i(\omega) - \mu) \right| \leq C(\omega)\sqrt{32(\log n + 1) \log\log(16(\log n + 1))},$$

which is $\tilde{\mathcal{O}}(1)$. That is, $\sum_{i \leq n} \iota_i(X_i - \mu) = \tilde{\mathcal{O}}(1)$ almost surely. Let us now show that

$$\left| \sum_{i \leq n} \tilde{\lambda}_{i,\alpha_{1,n}}(X_i - \mu) \right| = \tilde{\mathcal{O}}(1) \tag{14}$$

almost surely as well. For $i \geq m(\omega)$,

$$\sqrt{\frac{2\log(2/\alpha_{1,n})}{\widehat{\sigma}_i^2 i \log(i+1)}} \leq \frac{\sigma}{\sqrt{2}}\sqrt{\frac{\log(2/\alpha_{1,n})}{i \log(i+1)}}$$

and so, for $i \geq m(\omega) \vee \sigma\sqrt{\log(2/\alpha_{1,n})}c_5$, it holds that

$$\tilde{\lambda}_{i,\alpha_{1,n}} = \sqrt{\log(2/\alpha_{1,n})}\iota_i.$$

Given that we will let $n$ tend to $\infty$, we can assume without loss of generality that $m(\omega) < \sigma\sqrt{\log(2/\alpha_{1,n})}c_5 =: t_n$. In that

case,

$$
\begin{aligned}
\sum_{i \leq n} \tilde{\lambda}_{i,\alpha_{1,n}}(X_i - \mu) &= \sum_{i < t_n} \tilde{\lambda}_{i,\alpha_{1,n}}(X_i - \mu) + \sum_{i=t_n}^{n} \tilde{\lambda}_{i,\alpha_{1,n}}(X_i - \mu) \\
&= \sum_{i < t_n} \tilde{\lambda}_{i,\alpha_{1,n}}(X_i - \mu) + \sqrt{\log(2/\alpha_{1,n})} \sum_{i=t_n}^{n} \iota_i(X_i - \mu) \\
&= \sum_{i < t_n} \left( \tilde{\lambda}_{i,\alpha_{1,n}} - \sqrt{\log(2/\alpha_{1,n})}\iota_i \right) (X_i - \mu) \\
&\quad + \sqrt{\log(2/\alpha_{1,n})} \sum_{i \leq n} \iota_i(X_i - \mu).
\end{aligned}
$$

Now note that the absolute value of the first summand is upper bounded by

$$
\begin{aligned}
\sum_{i < t_n} \left| \tilde{\lambda}_{i,\alpha_{1,n}} - \sqrt{\log(2/\alpha_{1,n})}\iota_i \right| &\leq \sum_{i < t_n} \left| \tilde{\lambda}_{i,\alpha_{1,n}} - \sqrt{\log(2/\alpha_{1,n})}\iota_i \right| \\
&\leq \sum_{i < t_n} \left( c_5 + \sqrt{\log(2/\alpha_{1,n})} \sup_i \iota_i \right).
\end{aligned}
$$

Given that $\iota_n \to 0$ almost surely and $c_2 > 0$, $\sup_i \iota_i$ is almost surely bounded, and thus such a first summand is upper bounded by

$$
t_n \left( c_5 + \sqrt{\log(2/\alpha_{1,n})} \sup_i \iota_i \right),
$$

which is also $\tilde{\mathcal{O}}(1)$ a.s., in view of $\log(1/\alpha_{1,n}) = \tilde{\mathcal{O}}(1)$. Consequently, we have shown the validity of (14). Lastly, we observe that,

$$
\begin{aligned}
\sum_{i \leq n} \tilde{\lambda}_{i,\alpha_{1,n}} &= \sum_{i \leq n} \sqrt{\frac{2\log(2/\alpha_{1,n})}{\hat{\sigma}_t^2 i \log(1+i)}} \wedge c_5 \\
&\geq \frac{1}{\sqrt{\log(1+n)}} \sum_{i \leq n} \sqrt{\frac{2\log(2/\alpha)}{i}} \wedge c_5 \\
&\geq \frac{1}{\sqrt{\log(1+n)}} \left( \sqrt{2\log(2/\alpha)} \wedge c_5 \right) \sum_{i \leq n} \frac{1}{\sqrt{i}} \\
&\overset{(i)}{\geq} \frac{1}{\sqrt{\log(1+n)}} \left( \sqrt{2\log(2/\alpha)} \wedge c_5 \right) \left( 2\sqrt{n} - 2 \right),
\end{aligned}
$$

which is $\tilde{\Omega}\left(\sqrt{n}\right)$, where (i) is obtained in view of Lemma B.3. We thus conclude that

$$
|\hat{\mu}_t - \mu| = \left| \frac{\sum_{i=1}^{t-1} \tilde{\lambda}_i(X_i - \mu)}{\sum_{i=1}^{t-1} \tilde{\lambda}_i} \right| = \frac{\tilde{\mathcal{O}}(1)}{\tilde{\Omega}(\sqrt{t})} = \tilde{\mathcal{O}}\left(\frac{1}{\sqrt{t}}\right)
$$

almost surely.

### E.4. Proof of Proposition C.4

Given that $m_{4,n}^2 \to V\left[(X_i - \mu)^2\right]$ a.s. (in view of Proposition C.1), there exists $A \in \mathcal{F}$ such that $P(A) = 1$ and

$$
\hat{m}_{4,n}^2(\omega) \to V\left[(X_i - \mu)^2\right]
$$

for all $\omega \in A$. Based on Lemma B.12 and $c_2 > 0$,

$$\frac{1}{\widehat{m}_{4,n}^2(\omega)} \leq u(\omega) < \infty$$

for all $\omega \in A$. Given that $\delta_n \to \delta > 0$ and $\delta_n > 0$, then $l := \inf_n \delta_n > 0$, and so we observe that

$$\sqrt{\frac{2\log(1/\delta_n)}{\widehat{m}_{4,t}^2(\omega)n}} \leq \sqrt{\frac{2\log(1/l)}{u(\omega)n}},$$

which implies the existence of $m_\omega \in \mathbb{N}$ such that

$$\sqrt{\frac{2\log(1/l)}{u(\omega)n}} \leq c_1$$

for all $n \geq m_\omega$. Hence

$$\lambda_{t,\delta_n}(\omega) = \sqrt{\frac{2\log(1/\delta_n)}{\widehat{m}_{4,t}^2(\omega)n}}$$

for $n \geq m_\omega$. It follows that

$$\frac{1}{\sqrt{n}} \sum_{i=1}^n \lambda_{i,\delta_n}(\omega) = \frac{1}{\sqrt{n}} \sum_{i=1}^{m_\omega - 1} \lambda_{i,\delta_n}(\omega) + \frac{1}{\sqrt{n}} \sum_{i=m_\omega}^n \lambda_{i,\delta_n}(\omega).$$

Clearly, the first term converges to $0$, and so it suffices to show that

$$\frac{1}{\sqrt{n}} \sum_{i=m_\omega}^n \lambda_{i,\delta_n}(\omega) \stackrel{a.s.}{\to} \sqrt{\frac{2\log(1/\delta)}{\mathbb{V}\left[(X_i - \mu)^2\right]}}.$$

To see this, note that

$$\frac{1}{\sqrt{n}} \sum_{i=m_\omega}^n \lambda_{i,\delta_n}(\omega) = \frac{1}{\sqrt{n}} \sum_{i=m_\omega}^n \sqrt{\frac{2\log(1/\delta_n)}{\widehat{m}_{4,i}^2(\omega)n}}$$

$$= \underbrace{\frac{n - m_\omega}{n}}_{(I_n)} \underbrace{\frac{1}{n - m_\omega} \sum_{i=m_\omega}^n \sqrt{\frac{2\log(1/\delta_n)}{\widehat{m}_{4,i}^2(\omega)}}}_{(II_n(\omega))}.$$

Clearly, $(I_n) \stackrel{n \to \infty}{\to} 1$. Furthermore,

$$(II_n(\omega)) \stackrel{n \to \infty}{\to} \sqrt{\frac{2\log(1/\delta)}{\mathbb{V}\left[(X_i - \mu)^2\right]}}$$

in view of $m_{4,n}^2(\omega) \to V\left[(X_i - \mu)^2\right]$, $\delta_n \to \delta$, and Lemma B.8. Hence

$$\frac{1}{\sqrt{n}} \sum_{i=m_\omega}^n \lambda_{i,\delta_n}^{\mathrm{CI}}(\omega) \stackrel{n \to \infty}{\to} \sqrt{\frac{2\log(1/\delta)}{\mathbb{V}\left[(X_i - \mu)^2\right]}}$$

for $\omega \in A$, with $P(A) = 1$, thus concluding the proof.

### E.5. Proof of Proposition C.6

Analogously to the first part of the proof of Proposition C.4, there exists $A \in \mathcal{F}$ such that $P(A) = 1$,

$$\widehat{m}_{4,n}^2(\omega) \to V\left[(X_i - \mu)^2\right] \quad \forall \omega \in A, \quad \frac{1}{n}\sum_{i=1}^n Z_i(\omega) \to a \quad \forall \omega \in A, \tag{15}$$

and there exists $m_\omega$ such that

$$\lambda_{t,\delta_n}(\omega) = \sqrt{\frac{2\log(1/\delta_n)}{\widehat{m}_{4,t}^2(\omega)n}}$$

for $n \geq m_\omega$ for all $\omega \in A$. Observing that

$$\sum_{i=1}^n \psi_E\left(\lambda_{i,\delta_n}(\omega)\right)Z_i(\omega) = \underbrace{\sum_{i=1}^{m_\omega-1} \psi_E\left(\lambda_{i,\delta_n}(\omega)\right)Z_i(\omega)}_{(I_n(\omega))} + \underbrace{\sum_{i=m_\omega}^n \psi_E\left(\lambda_{i,\delta_n}(\omega)\right)Z_i(\omega)}_{(II_n(\omega))},$$

Clearly, $(I_n(\omega)) \to 0$ given that it is a linear combination of terms $\psi_E\left(\lambda_{i,\delta_n}(\omega)\right)$, with

$$\lambda_{i,\delta_n}(\omega) \overset{n\to\infty}{\searrow} 0, \quad \psi_E(\lambda) \overset{\lambda\to 0}{\to} 0.$$

Let us now prove that

$$(II_n(\omega)) \to \sqrt{\frac{2\log(1/\delta)}{\mathbb{V}\left[(X_i - \mu)^2\right]}}, \tag{16}$$

for $\omega \in A$. Denoting $\psi_N(\lambda) = \frac{\lambda^2}{2}$, as well as

$$\xi_{n,i}(\omega) := \frac{\psi_E\left(\lambda_{i,\delta_n}(\omega)\right)}{\psi_N\left(\lambda_{i,\delta_n}(\omega)\right)},$$

it follows that

$$\begin{aligned}
(II_n(\omega)) &= \sum_{i=m_\omega}^n \psi_E\left(\lambda_{i,\delta_n}(\omega)\right)Z_i(\omega) \\
&= \sum_{i=m_\omega}^n \psi_N\left(\lambda_{i,\delta_n}(\omega)\right)\xi_{n,i}(\omega)Z_i(\omega) \\
&= \frac{\log(1/\delta_n)(n - m_\omega + 1)}{n}\frac{1}{n - m_\omega + 1}\sum_{i=m_\omega}^n \frac{1}{\widehat{m}_{4,i}^2(\omega)}\xi_{n,i}(\omega)Z_i(\omega).
\end{aligned}$$

In view of (15), Lemma B.9 yields

$$\frac{1}{n - m_\omega + 1}\sum_{i=m_\omega}^n \frac{1}{\widehat{m}_{4,i}^2(\omega)}Z_i(\omega) \to \frac{a}{V\left[(X_i - \mu)^2\right]}.$$

Noting that $\frac{\psi_E(\lambda)}{\psi_N(\lambda)} \overset{\lambda\to 0}{\to} 1$, $\lambda_{i,\delta_i}(\omega) \geq \lambda_{i,\delta_n}(\omega)$ for $n \geq i$, and

$$\begin{aligned}
\lim_{n\to\infty}\lambda_{n,\delta_n}(\omega) &= \lim_{n\to\infty}\sqrt{\frac{2\log(1/\delta_n)}{\widehat{m}_{4,n}^2(\omega)n}} \\
&= \lim_{n\to\infty}\sqrt{\frac{1}{n}}\lim_{n\to\infty}\sqrt{\frac{2\log(1/\delta_n)}{\widehat{m}_{4,n}^2(\omega)}} \\
&= 0\sqrt{\frac{2\log(1/\delta)}{V\left[(X_i - \mu)^2\right]}} \\
&= 0,
\end{aligned}$$

we observe that

$$\xi_{n,n}(\omega) \to 1, \quad \xi_{i,i}(\omega) \ge \xi_{n,i}(\omega) \ge 1,$$

where the latter inequality follows from Lemma B.2. Invoking Lemma B.10 with

$$a_{n,i} = \xi_{n,i}(\omega), \quad b_i = \frac{1}{\widehat{m}_{4,i}^2(\omega)} Z_i(\omega),$$

it follows that

$$\frac{1}{n - m_\omega + 1} \sum_{i=m_\omega}^{n} \frac{1}{\widehat{m}_{4,i}^2(\omega)} \xi_{n,i}(\omega) Z_i(\omega) \to \frac{a}{V\left[(X_i - \mu)^2\right]}.$$

It suffices to observe that

$$\frac{\log(1/\delta_n)(n - m_\omega + 1)}{n} \to \log(1/\delta)$$

to conclude the proof.

### E.6. Proof of Proposition C.5

In view Proposition C.2 or Proposition C.3, there exists $A \in \mathcal{F}$ such that $P(A) = 1$ and

$$m_{4,t}^2(\omega) = \tilde{\mathcal{O}}\left(\frac{1}{t}\right) \tag{17}$$

for all $\omega \in A$. For $\omega \in A$, it may be that

$$\limsup_{t \to \infty} t m_{4,t}^2(\omega) =: M < \infty \tag{18}$$

or

$$\lim_{t \to \infty} t m_{4,t}^2(\omega) = \infty. \tag{19}$$

Denote $L := \sup_{n \in \mathbb{N}} \delta_n$, as well as $\kappa := \sqrt{\frac{2 \log(1/L)}{M}} \wedge c_1$. If (18) holds, then

$$
\begin{aligned}
\lambda_{t,\delta_n}(\omega) &= \sqrt{\frac{2 \log(1/\delta_n)}{\widehat{m}_{4,t}^2(\omega) n}} \wedge c_1 \\
&= \sqrt{\frac{2 \log(1/\delta_n)}{\widehat{m}_{4,t}^2(\omega) t} \frac{t}{n}} \wedge c_1 \\
&\ge \sqrt{\frac{2 \log(1/L)}{M} \frac{t}{n}} \wedge c_1 \\
&\overset{(i)}{\ge} \kappa \sqrt{\frac{t}{n}},
\end{aligned}
$$

where (i) follows from $\frac{t}{n} \le 1$. Thus

$$
\begin{aligned}
\frac{1}{\sqrt{n}} \sum_{i=1}^{n} \lambda_{i,\delta_n}(\omega) &\ge \frac{\kappa}{\sqrt{n}} \sum_{i=1}^{n} \sqrt{\frac{i}{n}} \\
&= \frac{\kappa}{n} \sum_{i=1}^{n} \sqrt{i} \\
&\overset{(i)}{\ge} \frac{2\kappa}{3n} n^{\frac{3}{2}} \\
&= \frac{2\kappa}{3} n^{\frac{1}{2}},
\end{aligned}
$$

where (i) follows from Lemma B.5.

If (19) holds, then there exists $m(\omega) \in \mathbb{N}$ such that, for $t \geq m(\omega)$,

$$\widehat{m}^2_{4,t} t \geq \frac{2\log(1/l)}{c_1^2},$$

where $l = \inf_{n \in \mathbb{N}} \delta_n$, which is strictly positive given that $\delta_n \to \delta > 0$ and $\delta_n > 0$. Thus

$$\sqrt{\frac{2\log(1/\delta_n)}{\widehat{m}^2_{4,t} n}} \leq \sqrt{\frac{2\log(1/l)}{\widehat{m}^2_{4,t} t}} \leq c_1,$$

and so

$$\lambda_{i,\delta_n}(\omega) = \sqrt{\frac{2\log(1/\delta_n)}{\widehat{m}^2_{4,t} n}}$$

for $i \geq m(\omega)$. It follows that

$$
\frac{1}{\frac{1}{\sqrt{n}} \sum_{i=1}^n \lambda_{i,\delta_n}(\omega)} \leq \frac{1}{\frac{1}{\sqrt{n}} \sum_{i=m(\omega)}^n \lambda_{i,\delta_n}(\omega)}
$$

$$
= \frac{n}{(n - m(\omega) + 1)\sqrt{2\log(1/\delta_n)}} \frac{n - m(\omega) + 1}{\sum_{i=m(\omega)}^n \frac{1}{\widehat{m}_{4,i}(\omega)}}
$$

$$
\overset{(i)}{\leq} \underbrace{\frac{n}{(n - m(\omega) + 1)\sqrt{2\log(1/\delta_n)}}}_{(I_n(\omega))} \underbrace{\sqrt{\frac{\sum_{i=m(\omega)}^n \widehat{m}^2_{4,i}(\omega)}{(n - m(\omega) + 1)}}}_{(II_n(\omega))},
$$

where (i) follows from the harmonic-quadratic means inequality. Now note that

$$(I_n(\omega)) \overset{n \to \infty}{\to} \sqrt{2\log(1/\delta)}.$$

In view of (17),

$$(II_n(\omega)) = \sqrt{\frac{\sum_{i=m(\omega)}^n \tilde{\mathcal{O}}\left(\frac{1}{i}\right)}{(n - m(\omega) + 1)}} = \tilde{\mathcal{O}}\left(\frac{1}{\sqrt{n}}\right),$$

We have shown that, regardless of (18) or (19) holding,

$$\frac{1}{\frac{1}{\sqrt{n}} \sum_{i=1}^n \lambda_{i,\delta_n}(\omega)} = \tilde{\mathcal{O}}\left(\frac{1}{\sqrt{n}}\right)$$

for all $\omega \in A$. Given that $P(A) = 1$, the proof is concluded.

### E.7. Proof of Proposition C.7

We will conclude the proof in two steps. First, we will prove that

$$(I_n) = \sup_{i \leq n} \left\{ \log(2/\alpha_{1,n}) + \sigma^2 \sum_{k=1}^{i-1} \psi_P\left(\sqrt{\frac{2\log(2/\alpha_{1,n})}{\widehat{\sigma}^2_k k \log(1+k)}} \wedge c_5\right) \right\}^2$$

scales polylogarithmically with $n$ almost surely. Second, we will show that

$$(II_n) = \sum_{i \leq n} \frac{1}{\left(\sum_{k=1}^{i-1} \sqrt{\frac{2\log(2/\alpha_{1,n})}{\widehat{\sigma}^2_k k \log(1+k)}} \wedge c_5\right)^2}$$

also scales polylogarithmically with $n$ almost surely. Thus, by Hölder's inequality and these two steps, it will follow that

$$\sum_{i \leq n} \frac{\left\{ \log(2/\alpha_{1,n}) + \sigma^2 \sum_{k=1}^{i-1} \psi_P \left( \sqrt{\frac{2 \log(2/\alpha_{1,n})}{\widehat{\sigma}_k^2 k \log(1+k)}} \wedge c_5 \right) \right\}^2}{\left( \sum_{k=1}^{i-1} \sqrt{\frac{2 \log(2/\alpha_{1,n})}{\widehat{\sigma}_k^2 k \log(1+k)}} \wedge c_5 \right)^2} \tag{20}$$

scales logarithmically with $n$ almost surely. That is, it is $\tilde{\mathcal{O}}(1)$ almost surely.

**Step 1.** If $\sigma = 0$, then $(I_n) = \log^2(2/\alpha_{1,n})$, which scales at most logarithmically with $n$ given that $1/\alpha_{1,n} = O(\log n)$, which follows from $\alpha_{1,n} = \Omega\left(\frac{1}{\log(n)}\right)$. If $\sigma > 0$, then in view of $(a+b)^2 \leq 2a^2 + 2b^2$, it follows that

$$(I_n) \leq \sup_{i \leq n} \left[ 2 \log^2(2/\alpha_{1,n}) + 2\sigma^4 \left\{ \sum_{k=1}^{i-1} \psi_P \left( \sqrt{\frac{2 \log(2/\alpha_{1,n})}{\widehat{\sigma}_k^2 k \log(1+k)}} \wedge c_5 \right) \right\}^2 \right].$$

We observe that

$$\psi_P \left( \sqrt{\frac{2 \log(2/\alpha_{1,n})}{\widehat{\sigma}_k^2 k \log(1+k)}} \wedge c_5 \right) \leq \psi_P \left( \sqrt{\frac{2 \log(2/\alpha_{1,n})}{\widehat{\sigma}_k^2 k \log(1+k)}} \right)$$

$$\stackrel{(i)}{\leq} \left( \frac{1}{\sqrt{2k \log(1+k)}} \right)^2 \psi_P \left( \sqrt{\frac{4 \log(2/\alpha_{1,n})}{\widehat{\sigma}_k^2}} \right)$$

$$= \frac{1}{2k \log(1+k)} \psi_P \left( \sqrt{\frac{4 \log(2/\alpha_{1,n})}{\widehat{\sigma}_k^2}} \right)$$

$$\stackrel{(ii)}{\leq} \frac{1}{2k \log(1+k)} \exp \left( \sqrt{\frac{4 \log(2/\alpha_{1,n})}{\widehat{\sigma}_k^2}} \right)$$

$$\leq \frac{1}{k} \exp \left( \sqrt{\frac{4 \log(2/\alpha_{1,n})}{\widehat{\sigma}_k^2}} \right)$$

$$\stackrel{(iii)}{\leq} \frac{1}{k} \exp \left( \frac{4 \log(2/\alpha_{1,n})}{\widehat{\sigma}_k^2} \right)$$

$$= \frac{1}{k} \left\{ \frac{2}{\alpha_{1,n}} \right\}^{\frac{4}{\widehat{\sigma}_k^2}},$$

where (i) follows from Lemma B.1 and $2k \log(1+k) \geq 1$ for all $k \geq 1$, (ii) follows from $\psi_P(x) = \exp(x) - x - 1 \leq \exp(x)$ for all $x \geq 0$, and (iii) follows from $\widehat{\sigma}_k \in [0, 1]$ and $\sqrt{x} \leq x$ for all $x \geq 1$.

Given Proposition C.1 and Lemma B.12 (in view of $c_3 > 0$), there exists $A \in \mathcal{F}$ such that $P(A) = 1$ and

$$\widehat{\sigma}_k^2(\omega) \to \sigma^2, \quad \inf_k \widehat{\sigma}_k(\omega) \geq \varkappa(\omega) > 0,$$

for all $\omega \in A$. For $\omega \in A$ and $k \in \mathbb{N}$,

$$\psi_P \left( \sqrt{\frac{2 \log(2/\alpha_{1,n})}{\widehat{\sigma}_k^2(\omega) k \log(1+k)}} \right) \leq \frac{1}{k} \left\{ \frac{2}{\alpha_{1,n}} \right\}^{\frac{4}{\varkappa^2(\omega)}},$$

and so

$$\sup_{i \leq n} \sum_{k=1}^{i-1} \psi_P \left( \sqrt{\frac{2 \log(2/\alpha_{1,n})}{\widehat{\sigma}_k^2(\omega) k \log(1+k)}} \right) \leq \left\{ \frac{2}{\alpha_{1,n}} \right\}^{\frac{4}{\varkappa^2(\omega)}} \sum_{k=1}^{i-1} \frac{1}{k}$$

$$\stackrel{(i)}{\leq} \left\{ \frac{2}{\alpha_{1,n}} \right\}^{\frac{4}{\varkappa^2(\omega)}} (\log i + 1)$$

$$\leq \left\{ \frac{2}{\alpha_{1,n}} \right\}^{\frac{4}{\varkappa^2(\omega)}} (\log n + 1),$$

where (i) is obtained in view of Lemma B.4. Thus

$$(I_n(\omega)) \leq 2 \log^2(2/\alpha_{1,n}) + \left\{ \frac{2}{\alpha_{1,n}} \right\}^{\frac{4}{\varkappa^2(\omega)}} (\log n + 1),$$

which scales polinomially with $\log n$ in view of $1/\alpha_{1,n} = O(\log n)$.

**Step 2.** Denoting $\kappa = \sqrt{4 \log(2/\alpha)} \wedge c_5$, it follows that

$$
\begin{aligned}
\sum_{k=1}^{i-1} \sqrt{\frac{2 \log(2/\alpha_{1,n})}{\widehat{\sigma}_k^2 k \log(1+k)}} \wedge c_5 &\geq \sum_{k=1}^{i-1} \sqrt{\frac{2 \log(2/\alpha)}{k \log(1+k)}} \wedge c_5 \\
&\overset{(i)}{\geq} \kappa \sum_{k=1}^{i-1} \sqrt{\frac{1}{2k \log(1+k)}} \\
&= \frac{\kappa}{\sqrt{2}} \sum_{k=1}^{i-1} \sqrt{\frac{1}{k \log(1+k)}} \\
&\geq \frac{\kappa}{\sqrt{2 \log(i)}} \sum_{k=1}^{i-1} \sqrt{\frac{1}{k}} \\
&\overset{(ii)}{\geq} \frac{2\kappa}{\sqrt{2 \log(i)}} \left[ (\sqrt{i-1} - 1) \vee 1 \right],
\end{aligned}
$$

where (i) follows from $2k \log(1+k) \geq 1$ for $k \geq 1$, and (ii) is obtained in view of Lemma B.3. It follows that

$$
\begin{aligned}
(II_n) &\leq \sum_{2 \leq i \leq n} \frac{\frac{2 \log(i)}{\kappa^2}}{\left[ (\sqrt{i-1} - 1) \vee 1 \right]^2} \\
&\leq \frac{2 \log(n)}{\kappa^2} \sum_{2 \leq i \leq n} \frac{1}{\left[ (\sqrt{i-1} - 1) \vee 1 \right]^2} \\
&= \frac{2 \log(n)}{\kappa^2} \left( 2 + \sum_{4 \leq i \leq n} \frac{1}{(\sqrt{i-1} - 1)^2} \right) \\
&\overset{(i)}{\leq} \frac{2 \log(n)}{\kappa^2} \left( 2 + \sum_{4 \leq i \leq n} \frac{9}{i} \right) \\
&\leq \frac{2 \log(n)}{\kappa^2} \left( 2 + \sum_{2 \leq i \leq n} \frac{9}{i} \right) \\
&\overset{(ii)}{\leq} \frac{2 \log(n)}{\kappa^2} (2 + 9 \log n),
\end{aligned}
$$

where (i) follows from $\sqrt{i-1} - 1 \geq \frac{\sqrt{i}}{3}$ for all $i \geq 4$, and (ii) follows from Lemma B.4. Thus, $(II_n)$ also scales polylogarithmically with $n$.

## F. Alternative approaches to the proposed empirical Bernstein inequality

We present in this appendix two alternative approaches to that proposed in Section 4.

### F.1. Decoupling the inequality into first and second moment inequalities

We start by presenting a naive approach to the problem using two empirical Bernstein inequalities, which may be the most natural starting point. However, this approach will prove suboptimal, both theoretically and empirically.

F.1.1. CONFIDENCE SEQUENCES OBTAINED USING TWO EMPIRICAL BERNSTEIN INEQUALITIES

We start by noting that

$$\sigma^2 = \mathbb{E}X_i^2 - \mathbb{E}^2 X_i,$$

Thus, in order to give an upper confidence sequence for $\sigma^2$, it suffices to derive an upper confidence sequence for $\mathbb{E}X_i^2$ and a lower confidence sequence for $\mathbb{E}X_i$. Consider

$$U_{1,\alpha_1,t} := \frac{\sum_{i \leq t} \lambda_i (X_i^2 - \widehat{m}_{2i})^2}{\sum_{i \leq t} \lambda_i} + \frac{\log(1/\alpha_1) + \sum_{i \leq t} \psi_E(\lambda_i) \left(X_i^2 - \widehat{m}_{2i}\right)^2}{\sum_{i \leq t} \lambda_i}$$

as the upper confidence sequence for $\mathbb{E}X_i^2$ (which follows from empirical Bernstein inequality), and

$$L_{2,\alpha_2,t} := \frac{\sum_{i \leq t} \tilde{\lambda}_i X_i}{\sum_{i \leq t} \tilde{\lambda}_i} - \frac{\log(1/\alpha_1) + \sum_{i \leq t} \psi_E(\tilde{\lambda}_i) \left(X_i - \widehat{\mu}_i\right)^2}{\sum_{i \leq t} \tilde{\lambda}_i}$$

as the lower confidence sequence for $\mathbb{E}X_i$ (which follows from empirical Bernstein), so that $\alpha_1 + \alpha_2 = \alpha$. Now we take

$$\sigma^2 \leq U_{1,\alpha_1,t} - L_{2,\alpha_2,t}^2 \tag{21}$$

as the upper confidence sequence for $\sigma^2$.

Similarly, in order to derive lower inequalities, define

$$L_{1,\alpha_1,t} := \frac{\sum_{i \leq t} \lambda_i (X_i^2 - \widehat{m}_{2i})^2}{\sum_{i \leq t} \lambda_i} - \frac{\log(1/\alpha_1) + \sum_{i \leq t} \psi_E(\lambda_i) \left(X_i^2 - \widehat{m}_{2i}\right)^2}{\sum_{i \leq t} \lambda_i}$$

as the lower confidence sequence for $\mathbb{E}X_i^2$ (which follows from empirical Bernstein inequality), and

$$L_{2,\alpha_2,t} := \frac{\sum_{i \leq t} \tilde{\lambda}_i X_i}{\sum_{i \leq t} \tilde{\lambda}_i} + \frac{\log(1/\alpha_1) + \sum_{i \leq t} \psi_E(\tilde{\lambda}_i) \left(X_i - \widehat{\mu}_i\right)^2}{\sum_{i \leq t} \tilde{\lambda}_i}$$

as the upper confidence sequence for $\mathbb{E}X_i$ (which follows from empirical Bernstein), so that $\alpha_1 + \alpha_2 = \alpha$. Now we take

$$\sigma^2 \geq L_{1,\alpha_1,t} - U_{2,\alpha_2,t}^2 \tag{22}$$

as the lower confidence sequence for $\sigma^2$.

F.1.2. THEORETICAL AND EMPIRICAL SUBOPTIMALITY OF THE APPROACH

Ideally, we would expect the width of the confidence interval for $\sigma^2$ to scale as $\sqrt{2\mathbb{V}(X-\mu)^2 \log(1/\alpha)/t}$ (i.e., first order term in Bennett's inequality). However, we see that the term in $U_{1,\alpha_1,t}$

$$\frac{\log(1/\alpha_1) + \sum_{i \leq t} \psi_E(\lambda_i) \left(X_i^2 - \widehat{m}_{2i}\right)^2}{\sum_{i \leq t} \lambda_i}$$

scales as $\sqrt{2\mathbb{V}X^2 \log(1/\alpha)/t}$. It suffices to observe that

$$\begin{aligned}
\mathbb{V}(X-\mu)^2 &= \mathbb{E}(X-\mu)^4 - \mathbb{E}^2(X-\mu)^2 \\
&= \mathbb{E}\left[X^4 - 4X^3\mu + 6X^2\mu^2 - 4X\mu^3 + \mu^4\right] - \left[\mathbb{E}^2 X^2 + \mu^4 - 2\mu^2 \mathbb{E}X^2\right] \\
&= \left(\mathbb{E}X^4 - \mathbb{E}^2 X^2\right) + \mathbb{E}\left[-4X^3\mu + 8X^2\mu^2 - 4X\mu^3\right] \\
&= \left(\mathbb{E}X^4 - \mathbb{E}^2 X^2\right) - 4\mu\mathbb{E}\left(X^{\frac{3}{2}} - X^{\frac{1}{2}}\mu\right)^2 \\
&= \mathbb{V}X^2 - 4\mu\mathbb{E}\left(X^{\frac{3}{2}} - X^{\frac{1}{2}}\mu\right)^2 \\
&\leq \mathbb{V}X^2,
\end{aligned}$$

where the last inequality follows from $\mu \in (0, 1)$, to conclude that the first order term of this confidence interval will generally dominate that of Bennett's inequality.

We also clearly see the suboptimality of the approach empirically. Figure 3 exhibits the upper and lower inequalities proposed in Section 4 to those derived in this appendix for all the scenarios considered in Section 5, illustrating the poor performance of the latter.

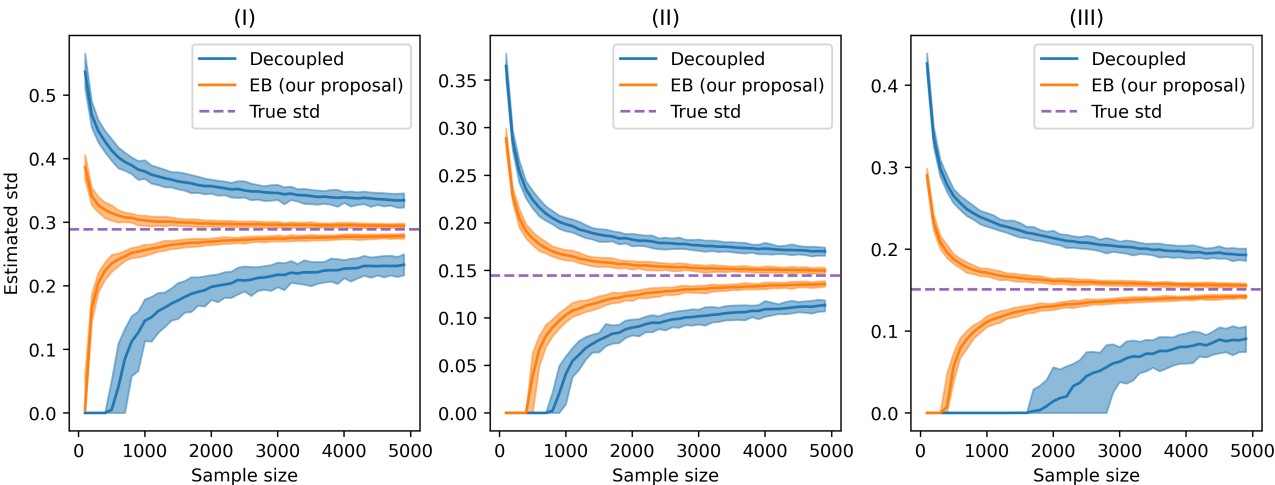

*Figure 3.* Average confidence intervals over 100 simulations for the std $\sigma$ for (I) the uniform distribution in $(0, 1)$, (II) the beta distribution with parameters $(2, 6)$, and (III) the beta distribution with parameters $(5, 5)$. For each of the inequalities, the $0.95\%$-empirical quantiles are also displayed. The decoupling approach (this appendix) is compared against EB (our proposal). EB clearly outperforms the decoupled approach in all the scenarios.

### F.2. Upper bounding the error term instead of taking negligible plug-ins

In Section 4.3, we proposed to take $\lambda_{t,l,\alpha_2} = 0$ if

$$\frac{\log(2/\alpha_1) + \hat{\sigma}_t^2 \sum_{i=1}^{t-1} \psi_P(\tilde{\lambda}_i)}{\sum_{i=1}^{t-1} \tilde{\lambda}_i} \leq 1.$$

A reasonable alternative would be to avoid defining $\lambda_{t,l,\alpha_2}$ as 0 (i.e., always define $\lambda_{t,l,\alpha_2} := \lambda_{t,u,\alpha_2}$), and to take

$$\tilde{R}_{t,\delta} = \begin{cases} \frac{\log(2/\delta) + \sigma^2 \sum_{i=1}^{t-1} \psi_P(\tilde{\lambda}_i)}{\sum_{i=1}^{t-1} \tilde{\lambda}_i}, & \text{if} \quad \frac{\log(2/\delta) + \hat{\sigma}_{t-1}^2 \sum_{i=1}^{t-1} \psi_P(\tilde{\lambda}_i)}{\sum_{i=1}^{t-1} \tilde{\lambda}_i} \leq 1, \\ 1, & \text{otherwise.} \end{cases}$$

In order to formalize this, denote

$$\Upsilon_t := \left\{ i \in [t] : \frac{\log(2/\delta) + \hat{\sigma}_{t-1}^2 \sum_{i=1}^{t-1} \psi_P(\tilde{\lambda}_i)}{\sum_{i=1}^{t-1} \tilde{\lambda}_i} \leq 1 \right\}, \quad \Upsilon_t^c := [t] \backslash \Upsilon_t.$$

Taking

$$A_t := \frac{\sum_{i \in \Upsilon_t} \lambda_i \tilde{A}_i}{\sum_{i \leq t} \lambda_i}, \quad B_{t,\delta} := 1 + \frac{\sum_{i \in \Upsilon_t} \lambda_i \tilde{B}_{i,\delta}}{\sum_{i \leq t} \lambda_i},$$

$$C_{t,\delta} := \frac{\sum_{i \in \Upsilon_t} \lambda_i \tilde{C}_{i,\delta} + \sum_{i \in \Upsilon_t^c} \lambda_i}{\sum_{i \leq t} \lambda_i},$$

in Section 4.3, Corollary 4.4 also holds. Figure 4 exhibits the empirical performance of this choice of plug-ins and that of Section 4.3, in the three scenarios from Section 5. The figure shows the slight advantage of considering the plug-ins from Section 4.3.

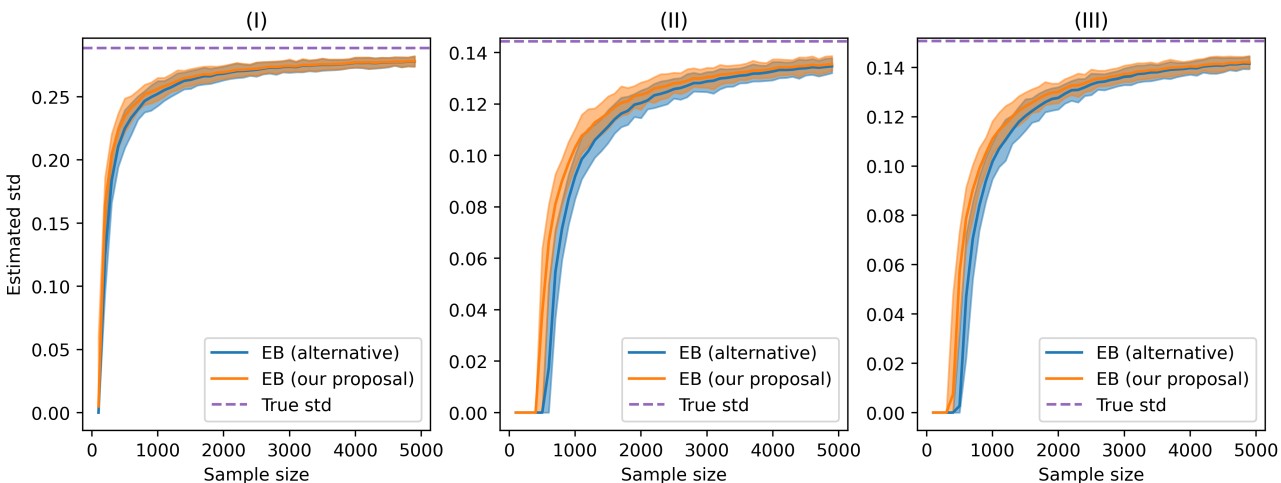

*Figure 4.* Average confidence intervals over 100 simulations for the std $\sigma$ for (I) the uniform distribution in $(0, 1)$, (II) the beta distribution with parameters $(2, 6)$, and (III) the beta distribution with parameters $(5, 5)$. For each of the inequalities, the $0.95\%$-empirical quantiles are also displayed. The EB lower confidence intervals with the plug-ins from Section 4.3 (our proposal) are compared against the EB lower confidence intervals with the plug-ins proposed in this appendix (alternative). Despite the expected similar outcomes, the plug-ins from Section 4.3 lead to slightly sharper bounds.

### F.3. Known mean

The main results of this contribution are derived from Corollary 4.2. However, Corollary 4.2 is not readily applicable because $E_t$ is unknown, given that $\mu$ is also unknown in practice. We explore here the power of Corollary 4.2 in comparison to our final confidence intervals, i.e., how much it is lost after dealing with the unknown term $E_t$. We explore this just as a theoretical exercise, given that $\mu$ is generally unknown. Figure 5 displays the upper and lower confidence intervals for different sample sizes and distributions. The upper confidence bounds remain essentially unchanged, with the orange and blue regions being nearly indistinguishable. In contrast, the lower bounds exhibit a clear gap, which is consistent with the greater difficulty of deriving lower bounds and the requirement of applying an additional concentration inequality to the empirical mean estimator.

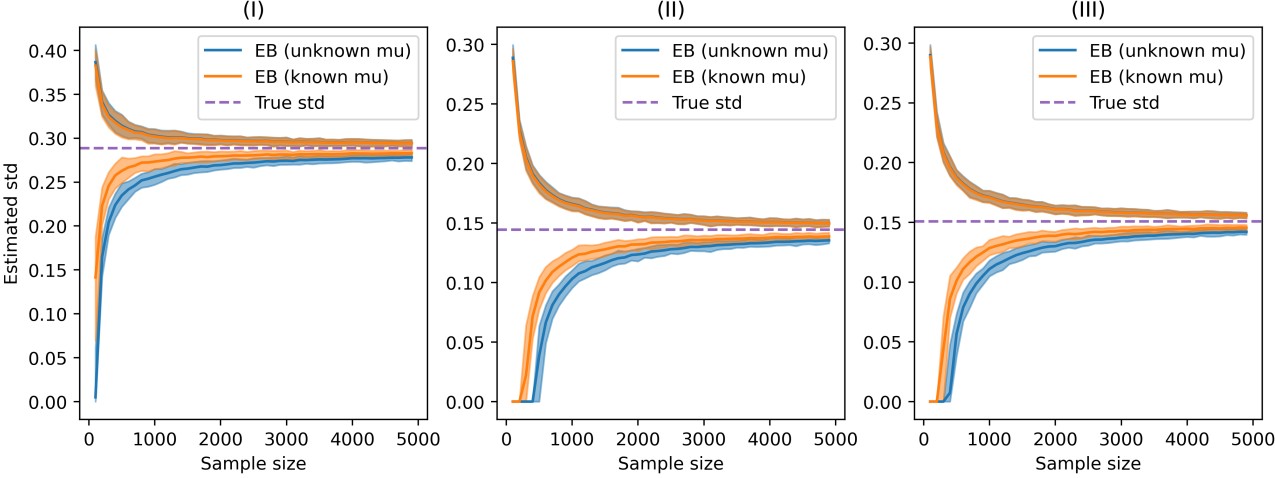

*Figure 5.* Average confidence intervals over 100 simulations for the std $\sigma$ for (I) the uniform distribution in $(0, 1)$, (II) the beta distribution with parameters $(2, 6)$, and (III) the beta distribution with parameters $(5, 5)$. The confidence intervals proposed in Section 4 (EB, unknown mu) are displayed alongside the confidence intervals that could be obtained from Corollary 4.2 if the mean $\mu$ was known (EB, known mu).

