# OpenReview forum: "Sharp Empirical Bernstein Inequalities for the Variance of Bounded Random Variables"
_ICML.cc/2026/Conference — ICML 2026 regular_

### Official Review · Reviewer_Wyxv · 2026-03-10

**Soundness:** 4
**Presentation:** 4
**Significance:** 3
**Originality:** 3
**Overall Recommendation:** 5
**Confidence:** 4

**Summary:**

This paper constructs a Bernstein-like concentration inequality for the variance of bounded random variables, under minimal assumptions on the data distribution (in particular: boundedness, constant conditional mean, and constant conditional variance). This inequality is shown to be sharp in the sense that the width of the corresponding confidence interval matches to the first term the width of the "oracle" Bernstein inequality confidence interval (constructed using knowledge of the moments of the distribution; see line 33). Both a batch version as well as an anytime-valid version (in the latter the sample size is not fixed ahead of time) of this inequality are established. Finally, numerical experiments are shown to validate the main theoretical results.

**Compliance With Llm Reviewing Policy:**

Affirmed.

**Final Justification:**

I'm happy to maintain my original score.

**Key Questions For Authors:**

See the strengths and weaknesses section.

**Limitations:**

Yes

**Strengths And Weaknesses:**

I think this is a good paper with a solid methodological contribution.

The paper fits into the recent line of work on anytime-valid inference, where confidence sequences are constructed via nonnegative supermartingales and Ville’s inequality. The presentation is extremely clear and well structured, which makes the technical arguments relatively easy to follow. The results appear correct to the best of my reading, and the proofs seem sound.
The authors establish asymptotic sharpness, with the first-order term in the width of the confidence interval matching the oracle Bernstein bound, which is an appealing optimality property. The paper also includes fair comparisons with existing bounds (most notably Maurer--Pontil), both theoretically and empirically.

The results seem potentially useful for applications, particularly in sequential decision-making settings where variance estimates play a role. One suggestion is that, since this contribution is primarily methodological and likely to be used as a tool in other applications, it would be helpful to expand the discussion of where such variance confidence sequences are most useful. The introduction briefly mentions some applications, but a more concrete discussion could strengthen the motivation. For instance, variance bounds can appear in settings such as high-probability conversion of regret bounds, or risk-sensitive decision-making. Elaborating on these connections would help readers better understand where the proposed results are likely to have the most impact.

Minor comment: on page 2, line 70, there appears to be missing or incomplete text.

---

> ### Author Rebuttal · Authors · 2026-03-28
>
> # Reviewer Wyxv
>
> We thank the reviewer for their consideration and insightful comments. We would like to address the following comments raised by the reviewer.
>
> - One suggestion is that, since this contribution is primarily methodological and likely to be used as a tool in other applications, it would be helpful to expand the discussion of where such variance confidence sequences are most useful. The introduction briefly mentions some applications, but a more concrete discussion could strengthen the motivation. For instance, variance bounds can appear in settings such as high-probability conversion of regret bounds, or risk-sensitive decision-making. Elaborating on these connections would help readers better understand where the proposed results are likely to have the most impact.
>
> We agree with the reviewer that the paper would benefit from an extended discussion on applications of variance confidence sequences. We have now included the following paragraph after introducing confidence sequences in the introduction.
>
> > Variance confidence sequences are particularly valuable when uncertainty quantification must be performed sequentially and in a data-dependent manner, and different options have different unknown variances. For instance, the optimal adaptive Neyman allocation in causal inference (Neopane et al., 2025) is based on confidence sequences for the variance. Similarly, in online learning and bandit problems, high-probability regret guarantees are often derived via concentration arguments, and sharp variance bounds enable tighter such conversions (Mnih et al., 2008; Audibert et al., 2009). Furthermore, in risk-sensitive decision-making, such as safe reinforcement learning or clinical trials (Kazerouni et al., 2017; Howard et al., 2021), sequential variance bounds allow practitioners to monitor and constrain risk in real time without relying on fixed horizons. More broadly, empirical Bernstein-type bounds play an important role in adaptive stopping rules, best-arm identification, and Monte Carlo estimation, where tighter variance estimates directly translate into improved sample efficiency.
>
> - Minor comment: on page 2, line 70, there appears to be missing or incomplete text.
>
> Corrected, thanks! It now reads as follows.
>
> > We further illustrate how our confidence sequences improve the adaptive Neyman allocation procedure from Neopane et al. (2025), which requires anytime-valid inference for the variance of the potential outcomes.

---

> > ### Author Rebuttal · Reviewer_Wyxv · 2026-04-03
> >
> > Thanks for the rebuttal. I maintain my positive review.

---

### Official Review · Reviewer_VPVU · 2026-03-11

**Soundness:** 3
**Presentation:** 2
**Significance:** 2
**Originality:** 3
**Overall Recommendation:** 3
**Confidence:** 5

**Summary:**

The paper studies empirical Bernstein inequalities for estimating the variance of sequence of bounded random variables under constant conditional expectation and variance. The authors derive non-asymptotic confidence intervals and confidence sequences for the variance in the offline and online setting. It is shown that their CI matches the benchmark-oracle Bernstein inequality asymptotically. Comparisions with other exisiting are made to show that their results yield an improvement.

**Compliance With Llm Reviewing Policy:**

Affirmed.

**Final Justification:**

Thanks to the authors for the rebuttal. Considering the empirical validity of the theoretical results and the presentation of the paper, I would prefer to maintain my score.

**Key Questions For Authors:**

1. Can you provide a sequential experiment where the anytime-valid nature is essential, such as optional stopping or adaptive monitoring, perhaps? An application-level improvement due to the results derived is essential to be shown.

2. How strong is the assumption in 4.5? Can one think of a weaker result without Assumption 4.5?

3. Potential typos and unclear parts:
Line 23, Column 2: denoted ---> denoted by
Line 32, Column 2, Equation: What are X, X_i? What do I and n denote? The equation and the paragraph above are a bit unclearly stated.
Page 2, Contributions bullet 3: The sentence is not complete.

**Limitations:**

yes

**Strengths And Weaknesses:**

Strengths:

1. It is an interesting problem, and finds applications in adaptive learning, policy evaluation, and bandits. The theoretical results seem non-trivial. The construction of the supermartingale in Theorem. 4.1. is the key novelty to develop interesting results.

2. The assumptions made to derive the results are weaker than the iid assumptions and a meaningful martingale-style generalization.

3. Matching the first-order width with the oracle Bernstein inequality is the right benchmark in my opinion, which depicts the sharpness of the results.

Weaknesses:

1. My major concern is about the fit of the paper to this venue. As presented, it is broadly a theoretical paper on concentration inequalities. It repeatedly mentions applications such as RL, bandits, off-policy evaluation, etc., but none of these applications are actually developed at all. I would prefer a stronger application section or a direct reduction showing how these results improve a concrete ML procedure.

2. The experiments are too narrow for the strength of the claims. It only compares against Maurer and Pontil on the simple bounded distributions. I would prefer more experiments to compare the derived results with other baselines and to show the robustness under non-iid processes satisfying the assumptions in the paper.

3. The presentation of the paper could have been better as it lacks clarity at certain places (more details below).

3. [Not a Weakness]: I did not fully verify all the proofs in the appendices, so my judgment is based on plausibility, structure, and clarity of arguments in the main paper.

---

> ### Author Rebuttal · Authors · 2026-03-28
>
> We thank the reviewer for their consideration and insightful comments. We would like to address the following points raised by the reviewer.
>
> The reviewer poses the following weakness:
>
> - As presented, the paper is broadly a theoretical paper on concentration inequalities. It repeatedly mentions applications, but none of these applications are actually developed at all.
>
> as well as the following question:
>
> - Can you provide a sequential experiment where the anytime-valid nature is essential, such as optional stopping or adaptive monitoring, perhaps? An application-level improvement due to the results derived is essential to be shown.
>
> We agree about the importance of providing a concrete application, in particular one that exploits the anytime-valid guarantees of our confidence sequences. We respectfully clarify that exactly such a detailed application to optimal adaptive Neyman allocation has already been provided in Section 5 (Experiments). The optimal adaptive Neyman allocation procedure sequentially allocates treatment assignments following a policy that is sequentially updated; in particular, this policy (derived in others' published work) is defined based on confidence sequences for the variance of the potential outcomes, and the procedure requires anytime validity inference on the variance in order to prove its optimality. Hence, this already employs **adaptive monitoring**, as requested. In Section 5, we demonstrate how our new confidence sequences translate into a reduction of the average mean square error by over $50\%$ with respect to an oracle procedure (see Figure 2).
>
>
> We now recognize the value of further highlighting this application, and we thank the reviewer for it. Hence, we have ensured that this connection is more prominently highlighted both in the introduction and experimental section to avoid any oversight by other readers. In particular, we have included the following comment in the last bullet point of the introduction (there was a typo before).
>
> > We further illustrate how our confidence sequences improve the adaptive Neyman allocation procedure from Neopane et al. (2025), which requires anytime-valid inference for the variance of the potential outcomes, but used inferior ones to ours.
>
> And we have also extended the following discussion in the experimental section (Section 5).
>
> > In Neopane et al. (2025), the probability of treatment assignment at a given time is calculated using confidence sequences for the standard deviations of the treatment and placebo outcomes. **In particular, the treatment policy is sequentially updated, and it requires anytime-valid inference on the variance to prove its optimality.** We conduct experiments using...
>
>
>
> The reviewer also poses the following question.
>
> - How strong is the assumption in 4.5? Can one think of a weaker result without Assumption 4.5?
>
> Assumption 4.5 is equivalent to having a constant (centered) fourth moment. We highlight that the concentration inequalities presented in this paper do not require this assumption in order to construct them or in order to be valid. This assumption is only made when mathematically studying the limiting width of the confidence interval, to see whether its first order limiting width matches that of the oracle Bernstein inequality (for the variance). Assumption 4.5 is really required for the oracle Bernstein to be sensibly interpreted as a comparison point. We have now  added the following comment after Assumption 4.5 to make this clear, which reads as follows.
>
> > We highlight that the concentration inequalities presented in this paper do not require Assumption 4.5 to be valid. We only impose Assumption 4.5 to compare the limiting width of our empirical inequalities to those of the oracle Bernstein inequality.
>
> - Potential typos and unclear parts: Line 23, Column 2: denoted ---> denoted by
>
> Corrected, thanks!
>
> - Line 32, Column 2, Equation: What are X, X_i? What do I and n denote? The equation and the paragraph above are a bit unclearly stated.
>
> We have now changed the second sentence of the paragraph to read as follows.
>
> > Assume for the moment that we observe $X_1, \ldots, X_n$, which are independent and identically distributed to $X$, which is a random variable taking values on [0,1] with mean $\mu$ and variance $\mathbb{V}(X)$.
>
>
>
> -  Page 2, Contributions bullet 3: The sentence is not complete.
>
> It now reads as discussed above, referencing the optimal Neyman allocation application.

---

> > ### Author Rebuttal · Reviewer_VPVU · 2026-04-03
> >
> > Thank you for the detailed rebuttal.

---

> > > ### Author Response · Authors · 2026-04-03
> > >
> > > Thank you again for the constructive comments, we are glad that our responses addressed all your concerns. As the discussion period is concluding, we would be happy to answer any remaining questions, otherwise we kindly request the reviewer to perhaps consider improving their score.

---

### Official Review · Reviewer_BScS · 2026-03-12

**Soundness:** 4
**Presentation:** 4
**Significance:** 3
**Originality:** 3
**Overall Recommendation:** 5
**Confidence:** 3

**Summary:**

The goal of this paper is to derive tight anytime confidence intervals for the
variance of a sequence of (potentially non-i.i.d, but with fixed conditional mean
and variance) bounded random variables.  Formally, we observe a sequence of
random variables $X_1, X_2, \dots, X_n, \dots$, taking values in the range $[0,
1]$.  These need not be independent, but we assume that at any time $t$,
conditioned on the "past", the conditional mean and variance of $X_{t}$ are
fixed to $\mu$ and $\sigma^2$, independent of the time $t$.  Formally, $X_{i}$
are adapted to a filtration $F_{i}$ such that $E[X_i|F_{i-1}]
= \mu$ and $Var[X_i\vert F_{i-1}] = E[X_i^2|F_{i}] - \mu^2 =
\sigma^{2}$, independent of $i$.

The goal is then is to produce a sequence of confidence intervals $C_i$
(s.t. $C_i$ can be computed based only on the observed values of $X_1, X_2,
\dots, X_{i}$) so that with probability at least $1 - \alpha$, it is true that
at *all* times $t$, $\sigma^2 \in C_t$.  With this order of quantifiers, the
confidence interval sequence (known as an "anytime" sequence) has many useful
properties: no union bound over time is required, and the confidence interval
remain valid even if time $t$ is chosen at random based on the observations
(i.e. a stopping time).


The main result of the paper is to provide a construction of such an anytime
confidence interval sequence for the setting of bounded r.v.s with fixed
conditional mean and variance (these fixed values are *not*, of course, known to
the algorithm generating the confidence sequence).  This is based on designing an
appropriate supermartingale of adjusted exponential moments, which has been the
method of choice in the field of anytime confidence intervals for the past
decade.

The paper also evaluated the empirical and theoretical performance of the
designed intervals.  Theoretically, this is done via an asymptotic (in the limit
$t\rightarrow \infty$) comparison of the width of the confidence intervals
against what is given by the fixed sample size Bennett inequality (which cannot
be used, as the paper points out, in a setting where a good estimate of the
first four moments of the $X_{i}$ is not available).  A theoretical comparison
with the empirical Bernstein inequality at fixed sample size obtained from
results of Maurer and Pontil, and the paper shows their width bound is
asymptotically better by a factor of $2$. (The theoretical comparison is done
under the further assumption that $Var[(X_i-\mu)^2|F_{i-1}]$
is also independent of $i$.)  The paper also shows empirical evidence of this
evidence based on synthetic data (Figure 1).


## Minor comments
- l. 321. Missing space after period before "As emphasized..."

**Compliance With Llm Reviewing Policy:**

Affirmed.

**Final Justification:**

My original review was already quite positive, and the only potential weakness of the paper might be (as stated in the review) that the methods are quite similar to previous uses of supermartingales for deriving anytime confidence bounds. However, I think the paper is still interesting.   In terms of correctness, as I said in my review, while I did not have the time to check all details of proofs, what I did check was correct.

**Key Questions For Authors:**

- p1, display in second column.  The description of the Bernstein confidence
  interval here seems to be inequality. If the $\pm$ symbol is interpreted
  simply to be a $+$, then what is written as $C_{n}$ seems to be actually the
  half-width of $C_{n}$ ($C_n$ itself should be centred around $\mathbb{V}(X)$
  anyway in the Bernstein setting) when computed according to Bennett's
  inequality (see, e.g. Theorem 3 in the cited paper of Maurer and Pontil).

**Limitations:**

Yes

**Strengths And Weaknesses:**

Strengths:

- The paper considers a crisp and important problem: as mentioned in the paper,
  variance estimation is an important primitive in several settings, and anytime
  confidence intervals that asymptotically match the finite fixed sample
  guarantees give perhaps the best possible solution to the problem. The paper
  achieves this.

- The paper is in general written very clearly, with a clear demarcation between
  known results and the new ideas.  While I have not checked all the proofs in
  detail (or the code provided in the supplementary zip file), what I have
  checked seems to be correct (except one point in the introduction that should
  perhaps be easily fixable, listed in questions for authors below.)

Weaknesses:

- Perhaps the main weakness of the paper might be originality compared to
  previous work, especially the paper of Waudby-Smith and Ramdas and the papers
  by Howard et al.  For example, the construction of the super-martingale in the
  proof of the main result Theorem 4.1 follows an outline quite similar to
  similar constructions for mean estimation in the above papers (in particular,
  as the paper itself points out, the proof of Theorem 4.1 is quite close to the
  proof of Waudby-Smith-Ramdas, Theorem 2; I think it is also quite close to the
  proof of a similar result (Theorem 4) in the cited paper of Howard et al.,
  Ann. Stat. 2021).

  Nevertheless, even after this construction, a bit of messaging is needed in
  order to correctly incorporate empirical estimates of the mean.

---

> ### Author Rebuttal · Authors · 2026-03-28
>
> We thank the reviewer for their consideration and insightful comments. We would like to address the following question posed by the reviewer.
>
>
>
> - p1, display in second column. The description of the Bernstein confidence interval here seems to be inequality. If the $\pm$ symbol is interpreted simply to be a $+$, then what is written as $C_n$ seems to be actually the half-width of $C_n$ ($C_n$ itself should be centred around $\mathbb{V}(X)$ anyway in the Bernstein setting) when computed according to Bennett's inequality (see, e.g. Theorem 3 in the cited paper of Maurer and Pontil).
>
> There is indeed a typo in the equation, we thank the reviewer for pointing it out. We have now corrected it, so it reads as
> > To recall, Bernstein's inequality for $\mathbb{X}$ implies that $\mathbb{P}(\mathbb{V}(X) \in C_n) \geq 1 - \alpha$ with $C_n = [D_n - R_n, D_n + R_n]$, where $D_n$ is the center of the confidence interval and $R_n$ is the radius $$R_n = \sqrt{\frac{2 \mathbb{V}[(X_i - \mu)^2] \log(2/\alpha)}{n}} + \frac{\log(2/\alpha)}{3n}.$$

---

> > ### Author Rebuttal · Reviewer_BScS · 2026-04-04
> >
> > Thank you for the response.  As my score was already positive I am keeping it as it is.

---

### Official Review · Reviewer_cs3x · 2026-03-13

**Soundness:** 4
**Presentation:** 3
**Significance:** 3
**Originality:** 4
**Overall Recommendation:** 5
**Confidence:** 3

**Summary:**

The paper gives the first sharp empirical Bernstein concentration inequality for the variance of the bounded variables. Given a stream of sampled random variables, the goal is to estimate the variance of the distribution. This work shows that under some standard assumptions (bounded variables, constant conditional mean, constant conditional variance), the empirical estimator concentrates well and they give sharp bounds for the same. They also compare it to the the previously best known estimator and show the improvement is theoretical as well as experimental.

**Compliance With Llm Reviewing Policy:**

Affirmed.

**Key Questions For Authors:**

1. Can you give justification for each of the assumptions you have made? Like which of them are really necessary and which of them can be relaxed without affecting the concentration bounds too much. And which of the assumptions actually hold in the applications you have mentioned?

**Limitations:**

Yes

**Strengths And Weaknesses:**

Strengths -

1. This work is technically sound and is more than incremental in nature, giving improved concentration bounds, improving over bounds known since a long time.
2. It is well motivated and has direct applications in bandit algorithms, reinforcement learning etc. That is an indicator as to the usefulness of the problem.
3. That they have experiments showing the improvements in such a technical area is very impressive. (Usually the theoretical improvements are hard to observe directly).
Weaknesses -
1. I feel that the writeup assumes a lot from the reader. The exposition could be improved to guide the reader through many parts of the proof.
2. I  am not an expert in the area, but the assumptions, though standard, seem to be very strong. A justification of each of the assumptions would be nice to have.

---

> ### Author Rebuttal · Authors · 2026-03-28
>
> We thank the reviewer for their consideration and insightful comments.
>
> The reviewer highlighted that the exposition of the proof could be improved by guiding the reader through more parts of the proof. We will use the extra page in the camera-ready version to improve on that end.
>
>
> We would also like to address the following question posed by the reviewer.
>
> -  Can you give justification for each of the assumptions you have made? Like which of them are really necessary and which of them can be relaxed without affecting the concentration bounds too much. And which of the assumptions actually hold in the applications you have mentioned?
>
> This is a very sensible point, and we realize that we should have initially discussed our Assumption 1.1 more extensively. We have now extended the paragraph after Assumption 1.1 to read as follows.
>
> > Note that any bounded random variable can be rescaled to belong to $[0,1]$, and so the first of the conditions can be assumed without loss of generality in the bounded setting. Since boundedness is a prerequisite for existing empirical Bernstein inequalities concerning the mean, it is unavoidable here too. Other than boundedness,  Assumption 1.1 remains substantially weak, replacing the traditional i.i.d. assumption with a broader martingale dependence structure. Furthermore, the conditional constant mean and variance are arguably the least we may assume if we wish to estimate "the variance". Since all bounded i.i.d. sequences can be rescaled to satisfy Assumption 1.1, our framework constitutes a strictly more general approach than the standard bounded i.i.d. assumptions prevalent in the literature. In particular, Assumption 1.1 is attained in all aforementioned applications, with some of them  requiring i.i.d. data (Maurer & Pontil, 2009; Austern & Mackey, 2022), and others requiring only martingale dependence (Howard et al., 2021; Neopane et al., 2025).
>
>
> It should now be clear that the assumptions made are rather minimal (there is no obvious way of relaxing them). Specifically, the requirement of boundedness is a standard prerequisite for the derivation of empirical Bernstein inequalities (which have only been derived for the mean thus far); thus, its inclusion should be viewed as a standard condition, rather than restrictive.
>
> Lastly, Assumption 4.5 is simply made in order to compare the limiting width of our confidence intervals with those from the oracle Bernstein inequality. Importantly, we do not require this assumption to hold for the proposed inequalities to be valid. We have further emphasized this in the manuscript; please see reply to reviewer VPVU.

---

> > ### Author Rebuttal · Reviewer_cs3x · 2026-04-03
> >
> > Thank you for the detailed answer.

---

### Decision · Program_Chairs · 2026-04-30

**Decision:**

Accept (regular)

**Comment:**

The paper gives a confidence interval for variance estimation of a sequence of bounded variables, improving upon a previous estimate of Marure-Pontil(2009). The improvement seems to be genuinely non-trivial and originates from a different method of proof (the new CI is much better for low-kurtosis data). However, as reviewers point out, this is not an ideal venue for such results, there no natural AI applications, and the work is largely just a standard martingale machinery (btw, what is with the name of Doob's maximal inequality in Thm 3.1??)  + solving a quadratic equation. Overall, however, paper does pass a bar in my batch, so I recommend acceptance.